# EDITLORD: Learning Code Transformation Rules for Code Editing

Weichen Li[1]  Albert Jan[2]  Baishakhi Ray[2]  Junfeng Yang[2]  Chengzhi Mao[3]  Kexin Pei[1]

## Abstract

Code editing is a foundational task in software development, where its effectiveness depends on whether it introduces desired code property changes without changing the original code's intended functionality. Existing approaches often formulate code editing as an implicit end-to-end task, omitting the fact that code editing procedures inherently consist of discrete and explicit steps. Thus, they suffer from suboptimal performance and lack of robustness and generalization. We introduce EDITLORD, a code editing framework that makes the code transformation steps explicit. Our key insight is to employ a language model (LM) as an inductive learner to extract code editing rules from the training code pairs as concise meta-rule sets. Such rule sets will be manifested for each training sample to augment them for finetuning or assist in prompting- and iterative-based code editing. EDITLORD outperforms the state-of-the-art by an average of 22.7% in editing performance and 58.1% in robustness while achieving 20.2% higher functional correctness across critical software engineering and security applications, LM models, and editing modes.

## 1. Introduction

Pre-trained code Language Models (code LMs) have shown impressive performance in automating software development and substantially improved the developers' productivity in a wide range of programming tasks (Shani & Staff, 2023). Among these tasks, code editing has been a fundamental building block with broad applications, such as optimizing code efficiency (Shypula et al., 2024; Huang et al., 2024; Peng et al., 2025b; Garg et al., 2022), reverse engineering and decompilation (Tan et al., 2024; Hu et al., 2024; Wong et al., 2023; Xie et al., 2024), and vulnerability

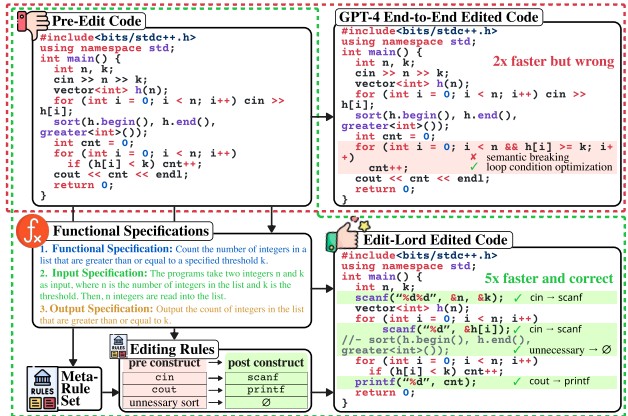

Figure 1: A performance editing example showing how ED-ITLORD differs from the existing approach (Shypula et al., 2024). The editing rule library here is discovered during the induction rule learning phase (Section 2.2) and used to prepare the finetuning set or directly prompting (Section 2.4). EDITLORD's edits include removing redundant sort and calling efficient I/O functions (cin to scanf), while Shypula et al. (2024) provides a suboptimal performance improvement while breaking the original code's input-output functionality.

repair and security hardening (Xia & Zhang, 2024; Fu et al., 2022; Peng et al., 2025a; He & Vechev, 2023; Perry et al., 2023; Bhatt et al., 2023).

Code editing is often considered more challenging than code generation, as it comes with extra requirements for introducing desired new properties, e.g., improved efficiency, in addition to generating functionally correct code. Such new requirements necessitate diagnosing and localizing specific properties in the original code, determining the necessary changes to edit these properties, and ensuring the edits do not introduce unintended side effects. These abstract editing procedures are often compositional because fulfilling certain code property changes can consist of multiple editing steps at multiple program points. However, they are also modular and reusable, as the same abstract editing steps can be reapplied in diverse code editing scenarios.

Unfortunately, existing works often treat the knowledge of code editing as latent and implicit, and resort to brute-force

[1]The University of Chicago [2]Columbia University [3]Rutgers University. Correspondence to: Weichen Li <weichenli@uchicago.edu>, Kexin Pei <kpei@uchicago.edu>.

*Proceedings of the 42nd International Conference on Machine Learning*, Vancouver, Canada. PMLR 267, 2025. Copyright 2025 by the author(s).

finetuning to internalize it as part of the model weights (Shypula et al., 2024; Tan et al., 2024). Such approaches could fail to decouple the modular code transformation procedures from the specific training samples and thus suffer from suboptimal performance (Section 3.2) and the lack of robustness and generalization (Section 3.5).

We introduce EDITLORD, a novel framework for learning explicit meta-program transformation rules for code editing tasks. Instead of directly prompting or finetuning an LM to perform the editing, EDITLORD first employs an LM to discover a concise set of inductive transformation rules from training data and then trains an LM to apply these transformations for each code pair. EDITLORD consists of three key subtasks: discovering edit rules to abstract the editing steps, summarizing functional specifications to maintain the code functionality, and learning the discovered rules for code editing. Figure 1 compares EDITLORD to the finetuned baseline (Shypula et al., 2024).

A key advantage of our approach is that it operates in a structured, discrete transformation space, reflecting the symbolic and compositional nature of code editing tasks. Since our discovered meta-program transformation rules are expressed in natural language, they remain interpretable. Moreover, by exposing the editing steps as explicit modules, EDITLORD enables more precise operations and allows human experts to intervene when necessary.

We evaluate EDITLORD on the three critical software engineering and security code editing tasks, including optimizing code efficiency, improving the readability of decompiled code, and repairing security vulnerabilities. EDITLORD outperforms the state-of-the-art code editing techniques by 23.3%, 12.7%, and 27.6%, respectively, across multiple code LMs. With explicit editing rules, EDITLORD significantly improved the baseline in generalization, e.g., by up to 24.87% improvement in length generalization and 58.1% improved robustness against semantics-preserving code transformations. In addition to finetuning, EDITLORD applies to different editing modes, with an average of 56.3% improvement in zero-shot prompting and 5.7% improvement for iterative refinement based on execution feedback. As EDITLORD exposes the explicit editing steps, we show that its editing performance can be further improved by up to 35.5% when steered by human experts.

## 2. Methodology

### 2.1. Problem Statement

The code editing task aims to transform a given pre-edit code into a post-edit code. The post-edit code must be semantically equivalent to the pre-edit code, i.e., preserving its input-output behavior and possessing the desired new code properties, e.g., improved efficiency, readability, or security (fewer vulnerabilities).

More formally, given a Language Model (LM) $M$ and a training dataset $\mathcal{D} = \{(x_i, y_i)\}_{i=1}^n, \mathcal{D} \subseteq X \times Y$, where $X = \{x_i\}_{i=1}^n$ is the set of pre-edit code samples and $Y = \{y_i\}_{i=1}^n$ is the set of post-edit code samples. Finetuning $M$ for code editing tasks can be formulated as optimizing the conditional probability $P_M(Y|X)$ with respect to $M$.

EDITLORD first creates an augmented training set $\mathcal{D}^\star$ before finetuning the model to perform rule-based code editing. Specifically, there are three steps: (1) data-driven inductive rule discovery, (2) functional specification discovery, and (3) rule-based code editing. Figure 2 illustrates the high-level workflow of EDITLORD. The first two steps bootstrap the training data $\mathcal{D}$, while the third step finetunes the code LM based on the augmented training set $\mathcal{D}^\star$.

In data-driven inductive rule discovery, we query an LM $G$ to iterate the training set and summarize a meta-rule set $R$. We will elaborate on the process in Section 2.2. In functional specification discovery, we extract the functional description $s_i \in S$ that describes the shared input-output behavior of both $x_i$ and $y_i$. Such a behavior is expected to be the same because the post-edit code $y_i$ should preserve the same input-output semantics of $x_i$. With the meta-rule set $R$ and the functional specifications $S$, we obtain an augmented training set $\mathcal{D}^\star \subseteq X \times Y \times R \times S$.

During finetuning, we train the LM $M$ to predict the per-sample editing rules $R_i$, functional specification $s_i$, before predicting the post-edit code $y_i$, all conditioned on the pre-edit code $x_i$: $P_M(y_i|x_i) = \sum_{s_i, R_i} P(s_i|x_i)P(R_i|x_i, s_i)P(y_i|x_i, s_i, R_i)$. Besides finetuning, we also consider other editing modes, e.g., prompting (Section 2.4).

### 2.2. Data-Driven Inductive Rule Discovery

Transformations required to achieve the desired changes in code property can be formalized as editing rules. Intuitively, these rules represent the explicit knowledge of code edits instead of the implicit weight internalized in the model. Our goal is to learn such an explicit meta-rule set $R$ from the training samples, before learning to perform the code editing guided by these explicit rules for better generalization. In the following, we describe how EDITLORD iterates the training set $\mathcal{D}$ to grow $R$ and ensures the rules in $R$ remain modular and reusable.

**Inductive meta-rule set initialization** Formally, we start by extracting raw editing rules by iterating each training code pair $(x_i, y_i)$. For each $(x_i, y_i)$, we query an LM $G$ to generate a per-sample raw editing rule set $R_i$ that explicitly describes the code property changes required to transform $x_i$ into $y_i$. Each raw rule set $R_i$ is thus defined as $R_i =$

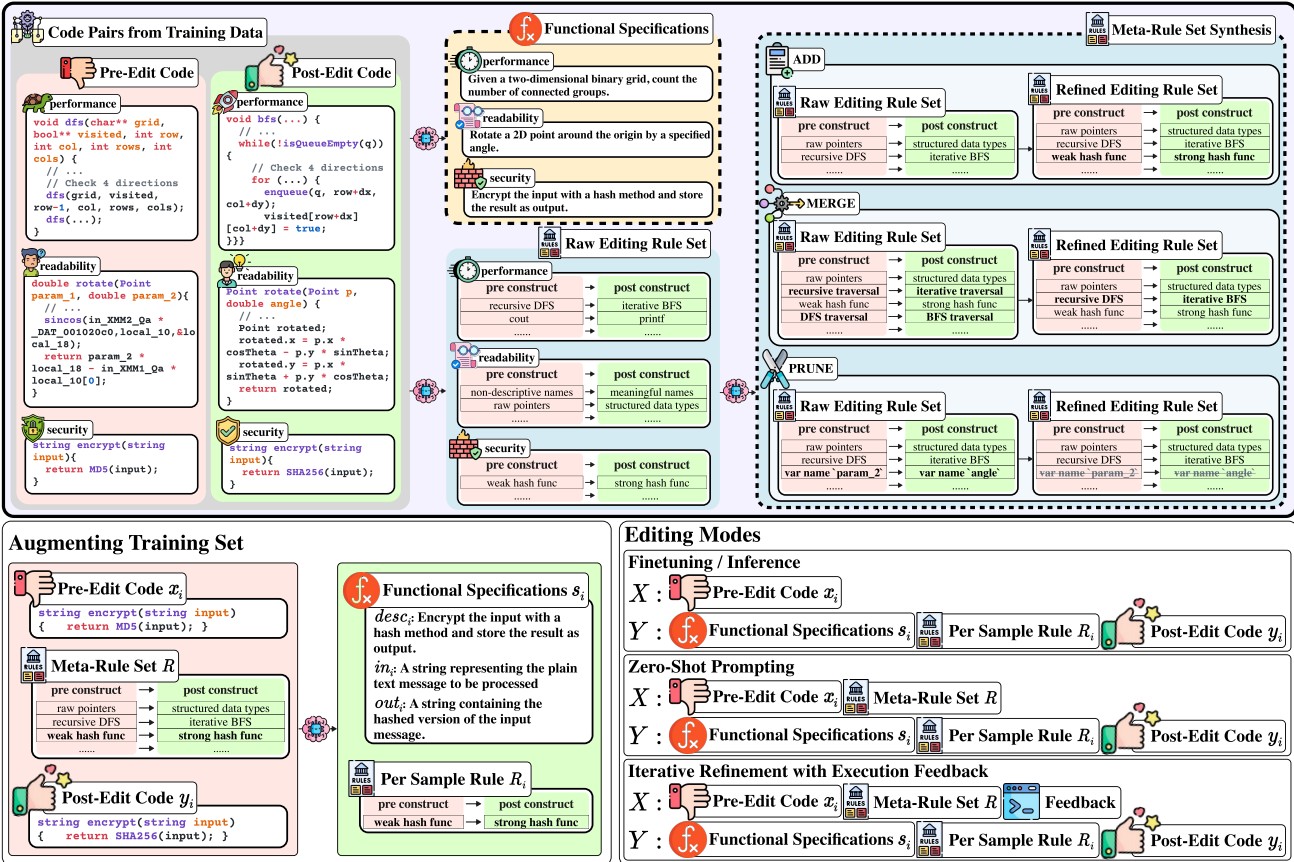

Figure 2: EDITLORD workflow. The upper section shows how EditLord takes pre-edit and post-edit training code samples as input and outputs the discovered meta-rule set and functional specifications (dotted-box) for each sample. The lower section shows how EDITLORD augment the training data. We finally augment each input with the synthesized functional specifications and per sample rule set derived from the meta-rule set. The editing process can then be performed either by querying the finetuned LMs (by augmented training samples) or using the meta-rule set as the prompt to guide the zero-shot prompting and the iterative refinement with external feedback. Note that the rule learning steps for three tasks are *independent*. We put them together just to show that the process is generic for different editing tasks.

$G(x_i, y_i)$. We then aggregate $\{R_1, ..., R_n\}$ to initialize the raw editing rule set $R = \bigcup_{i=1}^{n} R_i$.

**Iterative meta-rule set refinement** While the initial $R$ collects the raw editing rules for each sample, they are either too generic or too specific to provide actionable guidance for an effective editing. For example, they can be as generic as "check for potential vulnerability", or as specific as "switch from `a` to `a+1`" (with very specific variable names). Additionally, multiple editing rules may indicate similar transformations, e.g., "switch from `cin` to `scanf`" and "switch from `std::cin` to `scanf`". Given these challenges, we introduce an iterative meta-rule set refinement algorithm to ensure the ultimate $R$ is concise and effective. We start with this initial $R$ and iteratively refine each rule $r \in R$ until $R$ converges. Figure 2 illustrates the procedures.

To systematically perform the refinement, we define three operations that $G$ can use to update the inductive meta-rule set $R$.

- ADD: $R \cup \{r\} \to R$. This operation instructs $G$ to directly add the rule $r \in R_i$ derived from the training sample $(x_i, y_i)$ to the meta-rule set $R$. This step is still necessary as MERGE and PRUNE (described below) can be very aggressive and some useful rules have been inadvertently removed.

- MERGE: $R \cup \{r_i \oplus r_j\} \setminus \{r_i, r_j\} \to R$. This operation instructs $G$ to merge two rules $r_i, r_j \in R$ by replacing them with an updated rule $r_i \oplus r_j$.

- PRUNE: $R \setminus \{r\} \to R$. This operation instructs $G$ to remove the rule $r$ from the meta-rule set $R$.

**Algorithm 1** Iterative Meta-Rule Set Refinement
___
**input** Initial Meta-Rule Set $R$
**output** Finalized Meta-Rule Set $R'$
1: $R' \leftarrow R$
2: **while** $R'$ not converge **do**
3:    **for each** $r \in R \cup R'$ **do**
4:       **if** $r$ is well-balanced **then**
5:          **if** $\nexists r^* \in R'$ similar to $r$ **then**
6:             /* Add Rules */
7:             $R' \leftarrow R' \cup \{\, r \,\}$
8:          **else**
9:             /* Merge Rules */
10:            $R' \leftarrow R' \cup \{r \oplus r^*\} \setminus \{r, r^*\}$
11:          **end if**
12:       **else**
13:          /* Prune Rules */
14:          $R' \leftarrow R' \setminus \{\, r \,\}$
15:       **end if**
16:    **end for**
17: **end while**
18: **return** $R'$
___

Algorithm 1 shows the procedures of updating the editing rule set $R$. Before adding each $r$ (ADD and MERGE) to $R$, we prompt $G$ to assess whether it is a balanced rule, i.e., neither too generic nor too specific (line 4). If not, we apply PRUNE to discard it (line 14). Otherwise, we prompt $G$ again to decide whether we should ADD or MERGE $r$ (line 5). Specifically, $G$ will be prompted to decide whether there exists a rule in $R$ similar or identical to $r$. We apply MERGE (line 10) if yes and ADD (line 7) if not. Figure 2 and Table 5 include some examples of the learned rules in $R$ after this algorithm completes.

### 2.3. Functional Specification Discovery

We consider functional specification as a high-level natural language description of the program's intended behavior and input-output constraints. Specifically, for each sample $(x_i, y_i)$ in the training dataset, we define the functional specification $s_i$ of the training code pair $(x_i, y_i)$ as $s_i = \{desc_i, in_i, out_i\}$, where $desc_i$ describes the functionality implemented in $(x_i, y_i)$, and $in_i$ and $out_i$ specifies the input and expected output constraints of $(x_i, y_i)$, respectively. We prompt $G$ to generate $s_i$.

### 2.4. Code Editing Modes

**Finetuning for rule-based code editing**  Once we obtain the meta-rule set $R$, we iterate each training sample $(x_i, y_i)$, and prompt $G$ to identify $R_i \subseteq R$ as the per-sample editing rules that transform $x_i$ to $y_i$. With the per-sample rules $R_i$, we also incorporate the corresponding functional specifications $s_i$ to prepare the augmented finetuning set $D^\star = \{(x_i, s_i, R_i, y_i)\}_{i=1}^n$ (as shown in bottom-left in Figure 2).

The finetuning task can now be formulated as modeling the following conditional probability for each sample: $P_M(y_i, s_i, R_i | x_i) = P(s_i | x_i) P(R_i | x_i, s_i) P(y_i | x_i, s_i, R_i)$. Note that the LM $M$ to be finetuned may differ from $G$, which is used to construct the meta-rule set $R$ and functional specifications $S$ (see Section 3).

**Other editing modes**  Beyond editing the code using the finetuned model, the generic design of EDITLORD also supports other editing modes. For example, with the meta-rule set $R$, we can simply prompt an LM $M$ to directly generate the edited code or further refine the edited code based on external feedback (e.g., execution information (Peng et al., 2025b)). Figure 2 illustrates other code editing modes, i.e., zero-shot prompting and iterative refinement with execution feedback. We study how EDITLORD assists these additional editing modes in Section 3.4.

## 3. Experiments

We evaluate EDITLORD on three critical software engineering and security applications that can be formulated as code editing tasks (Section 3.1). We consider inference using the finetuned model as our default editing mode and compare it to the state-of-the-art baselines, which are also mostly based on finetuning (Shypula et al., 2024; Tan et al., 2024; Fu et al., 2022). In Section 3.4, we also show EDITLORD complements other editing modes. We choose models that can be full-parameter finetuned on our local hardware (2x4 Nvidia L40S GPUs), i.e., open-source DeepSeek-Coder 1.3B and 6.7B, or via an online API, i.e., GPT-4o mini. To generate the functional specifications and editing rules, we use GPT-4o mini ($G$ in Section 2).

### 3.1. Setup: Tasks, Datasets, and Metrics

**Performance optimization**  This task aims to edit a given program to improve its execution efficiency (see prompts in Appendix A.1). Following Shypula et al. (2024), we specifically focus on the execution time speedup and use gem5 CPU simulator (Binkert et al., 2011) to mitigate the noises introduced by the bare metal. We use the HQ (high quality) dataset from Shypula et al. (2024) for training and evaluation. The dataset consists of 4,085 training (slow and fast code) pairs, 2,544 validation samples, and 978 testing samples.

We adopt the same metrics as Shypula et al. (2024): *Correct@k*: the percentage of problems in the testing set for which the LM generates at least one correct solution out of the $k$ candidates. *OPT@k*: the percentage of problems in

```
Ghidra Decompiled Code
void * func0(
  char *param_1,
  int *param_2){ ✗ non-descriptive
  // ...
  if (cVar1 == '\0') {
    // ...
  }
  else {
    // ....
    do {
      // ....
      if (cVar1 == ')') {
        iVar3 = iVar3 + -1;
        if (iVar3 == 0) {
          iVar5 = iVar5 + 1;
          __ptr = realloc(__ptr,
(long)iVar5 * 4);
  ✗ raw pointer *(int *)((long)__ptr +
((long)iVar5 * 4 - 4U)) = iVar4;
          iVar4 = 0;
        }
      }
      cVar1 = *pcVar2;
      pcVar2 = pcVar2 + 1;
    } while (cVar1 != '\0');
  }
  *param_2 = iVar5;
  return __ptr;
}
```

```
Original Source Code
int* func0(
  const char* paren_string,
  int* returnSize) { ✓ meaningful
  // ...
  for (i = 0; paren_string[i]
!= '\0'; i++) {
    chr = paren_string[i];
    // ...
    if (chr == ')') {
      level -= 1;
      if (level == 0) {
        all_levels =
(int*)realloc(all_levels,
sizeof(int) * (count + 1));
        all_levels[count++] =
max_level;       ✓ structured type
        max_level = 0;
      }
    }
  }
  *returnSize = count;
  return all_levels;
}
```

```
Similarity Metric
Char based: 31.27
Token based: 33.43
Embedding based: 33.94
```

Figure 3: The similarity metrics between the Ghidra decompiled code and its corresponding source code are consistently low. This outcome is primarily due to the non-descriptive variable names and convoluted data structures, which result in both poor readability and consistently low similarity scores.

the testing set that the fastest and correct code among the $k$ LM generated programs is at least 10% faster. *Speedup@$k$*: the average absolute ratio between the execution time required by the given slow code and the fastest and correct code among the $k$ LM-generated programs.

**Decompilation** This task aims to edit a highly unreadable decompiled code into a more readable form (see prompts in Appendix A.2). We obtain these decompiled code samples using the off-the-shelf decompiler Ghidra (Agency, 2019). Figure 3 shows an example of the unreadable decompiled code and its original source code.

To construct training and validation samples, we follow Tan et al. (2024) to randomly sample original code snippets from AnghaBench (Da Silva et al., 2021) and construct the corresponding Ghidra decompiled code samples by compiling the original code and decompiling it. We keep the testing samples strictly non-overlapping with the training by using HumanEval-Decompile (Tan et al., 2024).

Ghidra decompiled code in HumanEval-Decompile may contain both syntactic and semantic errors, i.e., the pre-edit decompiled code may not compile or fail the test cases. These samples make it impossible for any code editor to reconstruct the semantically correct code without an oracle, which is often not available in typical code editing scenarios (Section 4). Therefore, we only consider the functionally correct decompiled code from the HumanEval-

Decompile (Tan et al., 2024). Overall, our dataset set consists of 8,567 (machine-decompiled code, original source code) training code pairs, 834 validation samples, and 131 testing samples with test cases.

We adopt the same metrics in Tan et al. (2024) and extend its readability measurements. *Compilability*: the percentage of problems in the testing set where the model generates the compilable program. *Correctness*: the percentage of problems in the testing set where the model generates the correct program. *Readability*: similarity between the ground truth and the edited code at three different levels, i.e., character-, token-, and embedding-level. The similarity is defined as $1 - d$, where $d$ is the edit distance between the ground truth source code and the recovered original code at the character and token level, and cosine distance between CodeSage (Zhang et al., 2024) embeddings of them at the embedding-level.

**Security hardening** This task aims to edit a given vulnerable code into the patched version (see prompts in Appendix A.3). In the vulnerable code, there can be one or more vulnerabilities under different Common Weakness Enumeration (CWE) categories.

We obtain the vulnerable and secure code pairs from SVEN (He & Vechev, 2023) for training and validation, and evaluate EDITLORD on a strictly unseen testing set, CWEval (Peng et al., 2025a). CWEval includes vulnerable code samples covering 31 CWEs. Importantly, we choose CWEval because each of its samples includes both functionality and security tests to automate the evaluation, while the other benchmarks, e.g., (Fu et al., 2022) focus only on security fixes and rely on manual effort to check the functionality.

Following CWEval (Peng et al., 2025a), we define $n$ as the total number of sampled solutions and a varying $k \leq n$. We then evaluate the security repair performance with the following three metrics. *Correct@$k$*: the expectation of any of $k$ LM generated solutions is correct. *Security@$k$*: the expectation of any of $k$ LM generated solutions is secure. *Correct & Security@$k$*: the expectation of any of $k$ LM generated solutions is both correct and secure.

These metrics are calculated through the same formula $\mathbb{E}_{\text{Problems}} \left[ 1 - \frac{\binom{n-c}{k}}{\binom{n}{k}} \right]$, where $c$ is the number of correct programs, secure programs, and both correct and secure programs, respectively.

### 3.2. Main Results

We compare EDITLORD to zero-shot prompting, chain-of-thought prompting (CoT), and finetuning (state-of-the-art baselines) on three tasks: performance optimization, de-

Table 1: Main results for performance optimization, decompilation, and security hardening. "Finetuned" refers to the approach from Shypula et al. (2024), Tan et al. (2024), Fu et al. (2022), respectively.

| Performance opt. | OPT@$k$ ↑ | | Speedup@$k$ ↑ | | Correct@$k$ ↑ | |
|---|---|---|---|---|---|---|
| | $k=1$ | $k=8$ | $k=1$ | $k=8$ | $k=1$ | $k=8$ |
| **DeepSeek-Coder 1.3B** | | | | | | |
| Prompt | 4.0 | 16.2 | 1.0 | 1.2 | 11.3 | 18.3 |
| CoT | 5.6 | 15.9 | 1.0 | 1.2 | 10.0 | 19.3 |
| Finetuned | 12.5 | 47.5 | 1.5 | 3.0 | 18.5 | 67.5 |
| EDITLORD (Ours) | **16.4** | **53.1** | **1.7** | **3.8** | **28.9** | **72.0** |
| **DeepSeek-Coder 6.7B** | | | | | | |
| Prompt | 6.0 | 19.5 | 1.1 | 1.2 | **60.0** | 89.5 |
| CoT | 4.1 | 6.8 | 1.1 | 1.2 | 16.0 | 36.6 |
| Finetuned | 7.8 | 76.9 | 2.1 | 3.1 | 20.7 | 87.4 |
| EDITLORD (Ours) | **11.0** | **82.5** | **2.7** | **4.5** | 36.0 | **89.5** |
| **GPT-4o mini** | | | | | | |
| Prompt | 12.2 | 26.4 | 1.1 | 1.3 | 47.4 | 71.5 |
| CoT | 10.0 | 29.1 | 1.1 | 1.5 | 45.2 | 64.1 |
| Finetuned | 28.1 | 61.5 | 2.3 | 3.8 | 59.0 | 89.5 |
| EDITLORD (Ours) | **31.2** | **72.5** | **2.9** | **4.2** | **62.5** | **93.5** |

| Decompilation | Compile↑ | Correct↑ | Readability↑ | | |
|---|---|---|---|---|---|
| | | | char | token | emb |
| **DeepSeek-Coder 1.3B** | | | | | |
| Prompt | 67.9 | **65.7** | 31.4 | 35.7 | 31.3 |
| CoT | 10.7 | 6.9 | 31.7 | 35.9 | 31.2 |
| Finetuned | 77.1 | 38.9 | 36.6 | 40.8 | 37.5 |
| EDITLORD (Ours) | **93.1** | 46.6 | **44.0** | **47.6** | **41.4** |
| **DeepSeek-Coder 6.7B** | | | | | |
| Prompt | 83.2 | **64.9** | 35.3 | 39.3 | 34.3 |
| CoT | 4.4 | 4.4 | 30.8 | 35.4 | 32.7 |
| Finetuned | 89.6 | 56.5 | 42.1 | 47.8 | 39.7 |
| EDITLORD (Ours) | **90.1** | 58.8 | **46.2** | **49.9** | **43.3** |
| **GPT-4o mini** | | | | | |
| Prompt | 61.8 | 44.3 | 33.1 | 37.0 | 37.9 |
| CoT | 59.5 | 46.6 | 34.3 | 38.3 | 39.8 |
| Finetuned | 86.3 | 52.7 | 45.6 | 49.4 | 43.6 |
| EDITLORD (Ours) | **97.7** | **67.2** | **51.4** | **54.9** | **48.4** |

| Security hardening | Correct@$k$ ↑ | | | Security@$k$ ↑ | | | Correct & Sec@$k$ ↑ | | |
|---|---|---|---|---|---|---|---|---|---|
| | $k=1$ | $k=10$ | $k=50$ | $k=1$ | $k=10$ | $k=50$ | $k=1$ | $k=10$ | $k=50$ |
| **DeepSeek-Coder 1.3B** | | | | | | | | | |
| Prompt | 6.7 | 26.8 | 40.4 | 3.0 | 15.2 | 30.8 | 1.0 | 7.1 | 15.4 |
| CoT | $21.2_{+14.5}$ | $39.7_{+12.9}$ | $48.1_{+7.7}$ | $6.4_{+3.4}$ | $20.2_{+5.0}$ | $34.6_{+3.8}$ | $1.8_{+0.8}$ | $9.4_{+2.3}$ | $17.3_{+1.9}$ |
| Finetuned | $24.1_{+17.4}$ | $35.7_{+8.9}$ | $40.4_{+0.0}$ | $9.5_{+6.6}$ | $21.5_{+6.3}$ | $28.9_{-1.9}$ | $5.3_{+4.3}$ | $13.8_{+6.8}$ | $19.2_{+3.8}$ |
| EDITLORD (Ours) | $\mathbf{31.2}_{+24.5}$ | $\mathbf{49.6}_{+22.8}$ | $\mathbf{59.6}_{+19.2}$ | $\mathbf{12.2}_{+9.2}$ | $\mathbf{28.4}_{+13.2}$ | $\mathbf{36.5}_{+5.8}$ | $\mathbf{7.4}_{+6.4}$ | $\mathbf{18.7}_{+11.6}$ | $\mathbf{23.1}_{+7.7}$ |
| **DeepSeek-Coder 6.7B** | | | | | | | | | |
| Prompt | 20.3 | 31.7 | 44.2 | 14.8 | 29.5 | 36.5 | 8.7 | 17.3 | 23.1 |
| CoT | $13.0_{-7.3}$ | $40.3_{+8.6}$ | $51.9_{+7.7}$ | $7.6_{-7.2}$ | $32.7_{+3.2}$ | $42.3_{+5.8}$ | $8.8_{+0.0}$ | $17.8_{+0.5}$ | $21.1_{-1.9}$ |
| Finetuned | $22.4_{+2.1}$ | $\mathbf{45.5}_{+13.8}$ | $\mathbf{53.9}_{+9.6}$ | $13.7_{-1.1}$ | $35.0_{+5.4}$ | $44.2_{+7.7}$ | $7.7_{-1.0}$ | $20.6_{+3.3}$ | $25.0_{+1.9}$ |
| EDITLORD (Ours) | $\mathbf{25.3}_{+5.0}$ | $41.3_{+9.6}$ | $49.1_{+4.9}$ | $\mathbf{17.9}_{+3.1}$ | $\mathbf{37.1}_{+7.6}$ | $\mathbf{50.0}_{+13.5}$ | $\mathbf{11.8}_{+3.1}$ | $\mathbf{23.9}_{+6.6}$ | $\mathbf{30.8}_{+7.7}$ |
| **GPT-4o mini** | | | | | | | | | |
| Prompt | 21.9 | 36.2 | 46.1 | 18.0 | 30.2 | 40.4 | 13.0 | 23.1 | 26.9 |
| CoT | $27.4_{+5.5}$ | $43.2_{+7.0}$ | $53.9_{+7.7}$ | $23.3_{+5.2}$ | $35.3_{+5.1}$ | $42.3_{+1.9}$ | $18.4_{+5.4}$ | $28.1_{+5.1}$ | $34.6_{+7.7}$ |
| Finetuned | $30.1_{+8.3}$ | $42.4_{+6.2}$ | $50.0_{+3.9}$ | $16.5_{-1.6}$ | $28.5_{-1.7}$ | $34.6_{-5.8}$ | $13.0_{+0.0}$ | $22.1_{-1.0}$ | $30.8_{+3.8}$ |
| EDITLORD (Ours) | $\mathbf{31.4}_{+9.5}$ | $\mathbf{49.9}_{+13.6}$ | $\mathbf{61.5}_{+15.4}$ | $\mathbf{27.6}_{+9.5}$ | $\mathbf{40.3}_{+10.1}$ | $\mathbf{50.0}_{+9.6}$ | $\mathbf{21.2}_{+8.2}$ | $\mathbf{29.3}_{+6.2}$ | $\mathbf{36.5}_{+9.6}$ |

compilation, and security hardening. As illustrated in Table 1, EDITLORD outperforms the state-of-the-art baselines across all tasks and models, with 23.3%, 12.7%, and 27.6% improvements on average in performance optimization, decompilation, and security hardening, respectively.

While EDITLORD underperforms the prompting-based approach based on DeepSeek-Coder 1.3B and 6.7B in the decompilation task and DeepSeek-Coder 6.7B in the performance task on the functional correctness, it is important to note that this metric alone can often appear overstated by simply keeping pre-edit code unmodified. The read-

ability metrics demonstrate that the prompting approach even *decreases* the original Ghidra decompiled code's readability, i.e., 36, 31.7, and 31.2 for the character-, token- and embedding-level readability metrics, respectively (Section 3.1).

### 3.3. Ablations

We conduct ablation studies to evaluate the effectiveness of each design component in EDITLORD. Specifically, we introduce the following variants by incrementally adding the key components (in both training and inference): (1)

Table 2: Ablations on performance optimization, decompilation, and security hardening (based on DeepSeek-Coder 1.3B).

| Performance opt. | OPT@k ↑ | | Speedup@k ↑ | | Correct@k ↑ | | Decompilation | Compile↑ | Correct↑ | Readability↑ | | |
|---|---|---|---|---|---|---|---|---|---|---|---|---|
| | $k=1$ | $k=8$ | $k=1$ | $k=8$ | $k=1$ | $k=8$ | | | | char | token | emb |
| Finetuned | 12.5 | 47.5 | 1.5 | 3.0 | 18.5 | 67.5 | Finetuned | 77.1 | 38.9 | 36.6 | 40.8 | 37.5 |
| + func-spec | 12.5 | 49.2 | 1.7 | 3.5 | 21.4 | **73.4** | + func-spec | **93.9** | 38.9 | 38.9 | 43.6 | 38.4 |
| + edit-rule | 15.7 | 51.9 | 1.8 | 3.7 | 26.2 | 70.0 | + edit-rule | 87.0 | 29.8 | 41.5 | 45.2 | 39.5 |
| EDITLORD (Ours) | **16.4** | **53.1** | **1.8** | **3.8** | **28.9** | 72.0 | EDITLORD (Ours) | 93.1 | **46.6** | **44.0** | **47.6** | **41.4** |

| Security hardening | Correct@k ↑ | | | Security@k ↑ | | | Correct and Sec@k ↑ | | |
|---|---|---|---|---|---|---|---|---|---|
| | $k=1$ | $k=10$ | $k=50$ | $k=1$ | $k=10$ | $k=50$ | $k=1$ | $k=10$ | $k=50$ |
| Finetuned | 24.1 | 35.7 | 40.4 | 9.5 | 21.5 | 28.9 | 5.3 | 13.8 | 19.2 |
| +func-spec | $25.5_{+1.3}$ | $41.8_{+6.1}$ | $46.1_{+5.8}$ | $11.5_{+1.9}$ | $27.3_{+5.8}$ | $36.5_{+7.7}$ | $6.0_{+0.7}$ | $15.8_{+2.0}$ | $21.1_{+1.9}$ |
| +edit-rule | $25.7_{+1.6}$ | $39.0_{+3.3}$ | $44.2_{+3.8}$ | $11.8_{+2.3}$ | $27.3_{+5.8}$ | $\mathbf{38.5_{+9.6}}$ | $6.7_{+1.3}$ | $17.5_{+3.6}$ | $23.1_{+3.8}$ |
| EDITLORD (Ours) | $\mathbf{31.2_{+7.1}}$ | $\mathbf{49.6_{+13.9}}$ | $\mathbf{59.6_{+19.2}}$ | $\mathbf{12.2_{+2.6}}$ | $\mathbf{28.4_{+6.9}}$ | $36.5_{+7.7}$ | $\mathbf{7.4_{+2.1}}$ | $\mathbf{18.7_{+4.8}}$ | $23.1_{+3.8}$ |

Table 3: Comparing EDITLORD to Tan et al. (2024) against semantics-preserving code transformations for robustness and unseen samples with longer lengths for generalization by measuring performance degradation with the original decompilation results (Table 1).

| | Compile↑ | Correct↑ | Readability↑ | | |
|---|---|---|---|---|---|
| | | | char | token | emb |
| Robustness | | | | | |
| Finetuned | 7.0 | 8.6 | 4.2 | 3.1 | 5.5 |
| EDITLORD (Ours) | **1.2** | **4.5** | **1.8** | **1.7** | **4.3** |
| Generalization | | | | | |
| Finetuned | 4.3 | 11.4 | 7.6 | 8.9 | 5.8 |
| EDITLORD (Ours) | **3.4** | **9.5** | **5.8** | **8.7** | **4.3** |

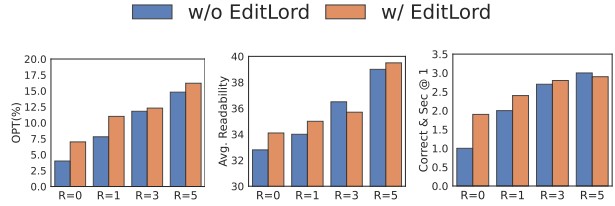

| w/o EditLord | w/ EditLord |

(a) Performance opt.  (b) Decompilation  (c) Security hardening

Figure 4: EDITLORD with different editing modes under zero-shot prompting ($R = 0$) and iterative refinement ($R = 1, 3, 5$) with execution feedback. Here, $R$ is the number of iterations taken to refine the edited code.

*Finetuned*: neither functional specification nor editing rule learning task is included; (2) *func-spec*: only the functional specification learning task is included; (3) *edit-rule*: only the editing rule learning task is included.

As shown in Table 2, adding the functional specification learning task alone outperforms the end-to-end finetuning baselines on the functional correctness metrics across all three tasks by up to 21.8%, while also improving the editing performance, e.g., efficiency, readability, and security, by up to 26.7%. Similarly, training models with the editing rule learning task alone can outperform the end-to-end finetuning baselines on the editing performance metrics across all three tasks by up to 33.2%, while improving the functional correctness by up to 17.1%. When combined, EDITLORD can outperform the end-to-end finetuning baselines by up to 34.8% in overall editing performance.

While EDITLORD underperforms these ablation variants on some specific metrics, i.e., *Correct@8* for performance,

*Compilability* for decompilation, and *security@50* for security hardening, EDITLORD obtains the best performance when considering both the functional correctness and editing performance.

### 3.4. Other Editing Modes

In addition to finetuning, EDITLORD is complementary to other code editing modes. We evaluate the effectiveness of EDITLORD when applied to zero-shot prompting and iterative refinement with execution feedback (Peng et al., 2025b; Huang et al., 2024).

As shown in Figure 4, EDITLORD improves the performance of zero-shot prompting and iterative refinement with execution feedback by an average of 56.3% and 5.7%, respectively. We observe EDITLORD underperforms slightly in the decompilation and security hardening task when $R = 3$ or $R = 5$. We suspect this is due to the coarse-grained execution feedback, e.g., compilation error, execution error, etc., while the state-of-the-art baselines incorporate more fine-grained information, e.g., per-statement execution profile (Huang et al., 2024) (see Section 4 for the

Table 4: Editing rules augmented by human experts using DeepSeek-Coder 1.3B.

| Performance opt. | OPT@$k \uparrow$ | | Speedup @$k \uparrow$ | | Correct@$k \uparrow$ | | Decompilation | Correct$_\uparrow$ | Compile$_\uparrow$ | Readability$_\uparrow$ | | |
|---|---|---|---|---|---|---|---|---|---|---|---|---|
| | $k=1$ | $k=8$ | $k=1$ | $k=8$ | $k=1$ | $k=8$ | | | | char | token | emb |
| EDITLORD | 16.4 | 53.1 | 1.7 | 3.8 | 28.9 | 72.0 | EDITLORD | 46.6 | **93.1** | 44.0 | 47.6 | 41.4 |
| + human | **17.9** | **61.4** | **1.9** | **3.9** | **53.0** | **76.9** | + human | **49.9** | 89.3 | **47.0** | **51.3** | **56.1** |

| Security hardening | Correct@$k \uparrow$ | | | Security@$k \uparrow$ | | | Correct & Sec@$k \uparrow$ | | |
|---|---|---|---|---|---|---|---|---|---|
| | $k=1$ | $k=10$ | $k=50$ | $k=1$ | $k=10$ | $k=50$ | $k=1$ | $k=10$ | $k=50$ |
| EDITLORD | 31.2 | 49.6 | 59.6 | 12.2 | 28.4 | 36.5 | 7.4 | 18.7 | 23.1 |
| + human | $\mathbf{33.2}_{+2.0}$ | $\mathbf{53.9}_{+4.3}$ | $\mathbf{65.4}_{+5.8}$ | $\mathbf{13.5}_{+1.3}$ | $\mathbf{33.8}_{+5.4}$ | $\mathbf{42.3}_{+5.8}$ | $\mathbf{8.9}_{+1.5}$ | $\mathbf{21.9}_{+3.2}$ | $\mathbf{28.9}_{+5.8}$ |

discussion).

### 3.5. Robustness and Generalization

**Robustness**  We compare EDITLORD to the state-of-the-art baseline against semantics-preserving code transformations (Wang et al., 2023a; Yefet et al., 2020; Gao et al., 2023; Yang et al., 2022; Bielik & Vechev, 2020). The transformations include renaming variables and functions to long random strings and removing comments, whitespaces, unused variables, and unused headers. We observe individual transformations alone rarely produce significant output changes. We thus exhaust each transformation (e.g., renaming all variables) and compose all of these transformations together to introduce input changes substantial enough.

Table 3 (left) shows the average performance degradation in functional correctness, comparability, and readability when the input undergoes semantics-preserving transformations. EDITLORD achieves up to 58.1% less drops in readability measured by character-level readability compared to the baseline, indicating that producing explicit functional specifications and editing rules helps EDITLORD stay less susceptible to syntactic code changes.

**Length generalization**  We investigate EDITLORD's generalizability to longer sequences than those seen in training (Anil et al., 2022). Specifically, we select 71 testing samples with lengths over 500 and finetune EDITLORD only on 6,105 strictly shorter ($< 500$) training samples.

Table 3 (right) demonstrates that EDITLORD suffers reduced performance degradation on unseen longer code, achieving up to 25.86% less drops in readability measured by embedding-level readability (Section 3.1).

### 3.6. Rule Augmentation by Human Experts

As EDITLORD produces explicit editing steps, it facilitates human intervention to introduce customized editing rules. To assess the effectiveness of EDITLORD augmented by human intervention, we incorporate two human experts (both

are the authors of this paper) to refine EDITLORD's generated editing rules and functional specifications for each testing sample, e.g., removing unreasonable rules or appending effective new rules.

Table 4 shows that such an augmentation enhances EDITLORD with up to 15.6%, 35.5%, and 25.1% improvement in efficiency, readability in decompilation, and security.

## 4. Discussion and Limitation

**Iterative refinement with execution feedback**  Existing LM-based code editing approaches often leverage iterative refinement with execution feedback (Huang et al., 2024; Peng et al., 2025b; Xia & Zhang, 2024; Waghjale et al., 2024), which relies on the availability of test inputs. However, the code to be edited may not always be well-maintained. Therefore, in this paper, we do not assume the tests are available by default (Section 3). We also show that EDITLORD is complementary to iterative refinement with coarse-grained execution feedback (Section 2.4). We aim to study whether the fine-grained execution feedback, i.e., per-statement execution profile (Huang et al., 2024), can further improve EDITLORD in the future.

**Functional correctness guarantee**  In code editing, functional correctness is arguably a hard constraint that cannot be violated, while the editing goals can be soft objectives. While we ensured that our editing performance is measured strictly on the subset of edited code that must first be functionally correct, EDITLORD cannot *guarantee* all the edited code is correct. While this issue can be trivially mitigated by falling back to the original code, it completely fails to introduce any edits. We thus follow the existing approaches (Shypula et al., 2024; Tan et al., 2024) by treating functional correctness as a soft constraint. Imposing a formal correctness guarantee for the edited code would be extremely valuable for future work.

## 5. Related Work

Code LMs have been extensively used to assist developers in editing code (Guo et al., 2025; Li et al., 2023; LaBash et al., 2024; Li et al., 2024; Cassano et al., 2024; Chakraborty et al., 2020; Liu et al., 2024a; Gupta et al., 2023; Muennighoff et al., 2023; Singhal et al., 2024) to fulfill various goals, such as bug fixing (Xia & Zhang, 2024; Fan et al., 2023; Liu et al., 2024c), decompilation (Tan et al., 2024; Wong et al., 2023; Xie et al., 2024; Hu et al., 2024), efficiency optimization (Huang et al., 2024; Shypula et al., 2024; Garg et al., 2022; Peng et al., 2025b), vulnerability repair (He & Vechev, 2023; Peng et al., 2025a; Fu et al., 2022; Xia & Zhang, 2024; Perry et al., 2023; Bhatt et al., 2023), code translation (Pan et al., 2024; Eniser et al., 2024), code refactoring (Shirafuji et al., 2023; Cummins et al., 2024), and more broadly related tasks like generating proofs (Chen et al., 2025; Chakraborty et al., 2025), invariants (Kamath et al., 2023; Chakraborty et al., 2023), or specifications (Murphy et al., 2024; Ma et al., 2025).

Most existing approaches either adopt finetuning to learn the direct mapping between the code pairs or leverage iterative refinement with execution or self-generated feedbacks (Huang et al., 2024; Peng et al., 2025b; Huang et al., 2023a; Xia & Zhang, 2024; Chen et al., 2024; Dong et al., 2024; Madaan et al., 2024; Zelikman et al., 2024; Liu et al., 2024b). In contrast, EDITLORD complements these approaches by making the intermediate editing steps explicit.

EDITLORD shares a similar philosophy to bootstrapping the symbolic reasoning of LMs (Zelikman et al., 2022; Kim et al., 2023; Huang et al., 2023b; Chen et al., 2024; Hsieh et al., 2023; Wang et al., 2023b; Zhang et al., 2023; Lightman et al., 2023; Zhou et al., 2024), where the LMs synthesize the reasoning procedures to self-augment the samples for supervised finetuning or preference tuning. EDITLORD extends the idea by distilling a concise and composable editing meta-rule set, such that the code to be edited can share reusable editing steps and benefits from the improved generalization (Section 3.5).

## 6. Conclusion

We introduced EDITLORD, a generic code editing framework, by learning the inductive code transformation rules to elicit the explicit code editing steps. Our key approach is to employ a language model (LM) as an inductive learner to distill a concise and composable meta-rule set for code editing from the training code pairs. EDITLORD substantially outperforms the state-of-the-art code editing techniques in editing performance while enjoying significantly higher functional correctness and improved robustness against semantics-preserving code transformations across multiple critical software engineering and security applications, LM models, and editing modes.

## Acknowledgement

We thank Xiaofei Ma, Jinjun Peng and all the anonymous reviewers for their constructive comments and feedback, which significantly improved this paper. We are also grateful to Jun Yang for his assistance with parts of the experimental setup. This work was supported in part by Columbia Center for AI Technology (in partnership with Amazon), NSF CCF 2313055, CCF 2107405, Chameleon (Keahey et al., 2020) and OpenAI Researcher Access Program (OpenAI).

## Impact Statement

Code editing plays a crucial role in assisting developers' daily jobs. While Large language models (LLMs) have demonstrated promising capabilities in automated code transformation, their end-to-end nature often leads to hallucinated edits. These hallucinations would be especially harmful when LLMs are used to edit security and safety-critical code. Our paper introduced a new approach to improving the safety of LLMs when applied to security-critical software engineering applications.

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

# A. Prompt Format

## A.1. Performance Optimization

---

**Prompt 1: Baseline prompt for performance optimization.**

**User:** This is the slow code:
[SLOW CODE]
{src_code}
[/SLOW CODE]
The corresponding fast code is:
[FAST CODE]
**Assistant:**{tgt_code}
[/FAST CODE]

---

**Prompt 2: Functional specifications only prompt for performance optimization.**

**User:** This is the slow code:
[SLOW CODE]
{src_code}
[/SLOW CODE]
The given code describes the following problem: {functional_specification}
The input specification is: {input_specification}
The output specification is: {output_specification}
The corresponding fast code is:
[FAST CODE]
**Assistant:**{tgt_code}
[/FAST CODE]

---

**Prompt 3: Editing rules only prompt for performance optimization.**

**User:** This is the slow code:
[SLOW CODE]
{src_code}
[/SLOW CODE]
Following editing rules should be applied: {editing_rules}
The corresponding fast code is:
[FAST CODE]
**Assistant:**{tgt_code}
[/FAST CODE]

---

**Prompt 4: Our prompt for performance optimization.**

**User:** This is the slow code:
[SLOW CODE]
{src_code}
[/SLOW CODE]
The given code describes the following problem: {functional_specification}
The input specification is: {input_specification}
The output specification is: {output_specification}
Following editing rules should be applied: {editing_rules}
The corresponding fast code is:
[FAST CODE]

---

**Assistant:**{tgt_code}
[/FAST CODE]

## A.2. Decompilation

Prompt 5: Baseline prompt for decompilation.

**User:** This is the decompiled code:
[MACHINE DECOMPILED CODE]
{src_code}
[/MACHINE DECOMPILED CODE]
The corresponding source code is:
[ORIGINAL SOURCE CODE]
**Assistant:**{tgt_code}
[/ORIGINAL SOURCE CODE]

Prompt 6: Functional specifications only prompt for decompilation optimization.

**User:** This is the decompiled code:
[MACHINE DECOMPILED CODE]
{src_code}
[/MACHINE DECOMPILED CODE]
The given code describes the following problem: {functional_specification}
The input specification is: {input_specification}
The output specification is: {output_specification}
The corresponding source code is:
[ORIGINAL SOURCE CODE]
**Assistant:**{tgt_code}
[/ORIGINAL SOURCE CODE]

Prompt 7: Editing rules only prompt for performance decompilation.

**User:** This is the decompiled code:
[MACHINE DECOMPILED CODE]
{src_code}
[/MACHINE DECOMPILED CODE]
Following editing rules should be applied: {editing_rules}
The corresponding source code is:
[ORIGINAL SOURCE CODE]
**Assistant:**{tgt_code}
[/ORIGINAL SOURCE CODE]

Prompt 8: Our prompt for decompilation.

**User:** This is the decompiled code:
[MACHINE DECOMPILED CODE]
{src_code}
[/MACHINE DECOMPILED CODE]
The given code describes the following problem: {functional_specification}
The input specification is: {input_specification}
The output specification is: {output_specification}

Following editing rules should be applied: {editing_rules}
The corresponding source code is:
[ORIGINAL SOURCE CODE]
**Assistant:**{tgt_code}
[/ORIGINAL SOURCE CODE]

## A.3. Security

Prompt 9: Baseline prompt for security.

**User:** This is the vulnerable code:
[VULNERABLE CODE]
{src_code}
[/VULNERABLE CODE]
The corresponding secure code is:
[SECURE CODE]
**Assistant:**{tgt_code}
[/SECURE CODE]

Prompt 10: Functional specifications only prompt for security.

**User:** This is the vulnerable code:
[VULNERABLE CODE]
{src_code}
[/VULNERABLE CODE]
The given code describes the following problem: {functional_specification}
The input specification is: {input_specification}
The output specification is: {output_specification}
The corresponding secure code is:
[SECURE CODE]
**Assistant:**{tgt_code}
[/SECURE CODE]

Prompt 11: Editing rules only prompt for security.

**User:** This is the vulnerable code:
[VULNERABLE CODE]
{src_code}
[/VULNERABLE CODE]
Following editing rules should be applied: {editing_rules}
The corresponding secure code is:
[SECURE CODE]
**Assistant:**{tgt_code}
[/SECURE CODE]

Prompt 12: Our prompt for security.

**User:** This is the vulnerable code:
[VULNERABLE CODE]
{src_code}
[/VULNERABLE CODE]

The given code describes the following problem: {functional_specification}
The input specification is: {input_specification}
The output specification is: {output_specification}
Following editing rules should be applied: {editing_rules}
The corresponding secure code is:
[SECURE CODE]
**Assistant:**{tgt_code}
[/SECURE CODE]

## A.4. Meta-Rule Set

Prompt 13: Generic/specific evaluation for an editing rule

Please analyze the provided editing rule (in order to improve {task_name}) and determine whether it is broadly applicable across different code snippets (generic) or tailored to a specific code snippet (specific). An editing rule like "{generic_rule_example}" should be considered as a generic rule. While a rule like "{specific_rule_example}" should be considered as a specific rule.
Provide your response in the following format:
The rule is [generic/specific] because ...
So, what do you think about the rule "{editing_rule}"? Is it generic or specific?

Prompt 14: Add/Merge rules

Please analyze the provided editing rule (in order to improve {task_name}) and compare it with the existing editing rules in the meta-rule set. If it's similar to any existing editing rule, please suggest how it should be integrated into the existing meta-rule set. Specify the one and only one appropriate action from the options below:
[ADD]: If none of the existing editing rules in the meta-rule set is similar to the current one, provide the refined and updated editing rule to be added to the set.
[MERGE]: If the current editing rule is similar to an existing editing rule, indicate which existing meta-rule is similar to the current editing rule so that they can be merged and how they should be merged.
If [ADD] is selected, please provide the refined and updated editing rule to be added to the set directly without any other information. If [MERGE] is selected, please provide exactly the existing meta-rule that is similar to the current editing rule with an updated editing rule.
Please notice that whether you select [ADD] or [MERGE], the editing rule you add or merge into the meta-rule set must adhere to the format "switch from ... to ...". Ensure that you only provide editing rules that transition from a {old_property} to {new_property}.
Here are several examples of the output:
[Example Output 1]
[ADD] only the editing rule to be added here [/ADD]
[/Example Output 1]
[Example Output 2]
[MERGE] only the editing rule to be merged and the updated rule, split by semicolon [/MERGE]
[/Example Output 2]
Meta-Rule Set:
{meta_rule_set}
Editing Rule Requested for Analysis:
{editing_rule}

## B. Meta-Rule Set Learned by EDITLORD

Table 5 shows some examples of meta-rules for code editing discovered by EDITLORD. We also calculate the percentage of edited samples (as indicated in the third column) that benefit from the specific rule in each row. For example, the rule

Table 5: Meta-rule examples discovered by EDITLORD for performance optimization, decompilation, and security hardening (Section 2.2). Each rule follows the universal format "switch *from* [old properties] *to* [new properties]". The third column shows the percentage of the samples improved by the specific rule.

| From | To | %↑ | Examples |
|---|---|---|---|
| **Performance Optimization** | | | |
| `cout` | `printf` | 32.4 | Figure 6 |
| `cin` | `scanf` | 24.8 | Figure 6 |
| multiple nested loops for condition checks | a single streamlined loop for condition checks and assignments | 21.0 | Figure 5 |
| recursive function calls | optimized iterative data handling methods | 7.6 | Figure 5 |
| dynamic memory allocation | static memory allocation | 7.6 | Figure 6 |
| **Decompilation** | | | |
| complex if-else structure | simplified conditional logic | 11.9 | Figure 7 |
| complex pointer arithmetic | clear variable assignments | 10.1 | Figure 7, 8 |
| redundant checks | straightforward boolean comparisons | 6.0 | Figure 8 |
| cryptic variable names | descriptive variable names | 5.9 | Figure 7 |
| ambiguous function signatures | clear function signatures | 5.9 | Figure 8 |
| **Security Hardening** | | | |
| no checks on function return values | check function return value | 32.9 | Figure 9 |
| direct SQL string interpolation | use of parameterized logic | 10.6 | Figure 10 |
| unvalidated input handling | check for buffer overflows on memory accesses | 8.9 | Figure 11 |
| unvalidated memory allocation | check maximum buffer allocation size before allocation | 7.6 | Figure 12 |
| direct parsing | implement comprehensive validation checks for character handling | 6.8 | Figure 13 |

"switch from `cout` to `printf`" applies to 32.4% of the testing samples that have obtained performance improvement.

# C. Case Study

## C.1. Performance Optimization

We show several examples to demonstrate how EDITLORD improve performance optimization.

Given slow code

```cpp
#include <bits/stdc++.h>

using namespace std;

long long n;long long k;
vector<long long> vec;
long long meme[100007];

long long solve(long long index){
  if(index==n-1){
    return 0;
  }
  if(meme[index]!=-1){
    return meme[index];
  }
  long long ret = 1000000000000000;
  long long sol = 1000000000000000;
  for(long long i = 1 ;i<=k ; i++){
    if(index+i<n){
      ret = abs (vec[index]-
      ↪ vec[index+i]) +
      ↪ solve(index+i);
    }
    sol = min(sol,ret);
  }
  return meme[index] = sol;
}

int main(){
  cin>>n;
  cin>>k;
  memset(meme,-1,sizeof(meme));
  long long num;
  for(long long i = 0 ; i<n ; i++){
    cin>>num;
    vec.push_back(num);
  }
  cout<<solve(0)<<endl;
  return 0;
}
```

EDITLORD output

```
The given code describe the following
↪ problem: Find the minimum cost to reach
↪ the last index of a sequence of integers,
↪ where the cost is defined as the absolute
↪ difference between consecutive integers.
The input specification is: The programs take
↪ two integers n and k as input, where n is
↪ the number of integers in the sequence
↪ and k is the maximum jump length. Then, n
↪ integers are read, representing the
↪ sequence.
The output specification is: Output the
↪ minimum cost to reach the last index from
↪ the first index.
Following editing rules should be applied:
switch from multiple nested loops to single
↪ linear iterations
switch from recursive function calls to
↪ optimized iterative data handling methods
The corresponding fast code is:
[FAST CODE]
```
```cpp
#include<bits/stdc++.h>

using namespace std;

long long int dp[100007],a[100007];

int main(){
  long long int n,k,i,j,x,y,z;
  cin>>n>>k;
  for(i=0;i<n;i++){
    cin>>a[i];
  }
  dp[0]=0;
  for(i=1;i<n;i++){
    x=10000000000007;
    for(j=i-1;j>=i-k&&j>=0;j--){
      x=min(x,dp[j]+abs(a[i]-a[j]));
    }
    dp[i]=x;
  }
  cout<<dp[n-1];
}
[/FAST CODE]
```

Figure 5: Performance optimization example 1.

Given slow code

```cpp
#include <bits/stdc++.h>

using namespace std;

int n ,k;
vector<int>vi;
long long mem[200005];
bool vis[200005];

long long solve(int indx ){
  if(indx == n-1)
    return 0;
  if(vis[indx])
    return mem[indx];
  vis[indx] = true;
  mem[indx] = 1e13;
  for(int i = indx+1; i <=min(indx+k
  ↪  , n-1) ; i++){
    mem[indx] = min(mem[indx] ,
    ↪  solve(i ) + abs(vi[indx] -
    ↪  vi[i]));
  }
  return mem[indx];
}

int main(){
  cin>>n>>k;
  vi.resize(n);
  for(int i = 0; i <n ; i++)
    cin>>vi[i];
  cout<<solve(0);
  return 0;
}
```

EDITLORD output

```
The given code describe the following
↪  problem: Calculate the minimum cost to
↪  reach the last index in a list of
↪  integers, where each step can jump up to
↪  k positions.
The input specification is: The programs take
↪  two integers n and k as input, where n is
↪  the number of integers in the list and k
↪  is the maximum jump length. Then, n
↪  integers are read, representing the cost
↪  associated with each index.
The output specification is: Output the
↪  minimum cost to reach the last index from
↪  the first index.
Following editing rules should be applied:
switch from multiple nested loops for
↪  counting elements to direct arithmetic
↪  calculation
switch from multiple nested loops to a
↪  compact function
switch from cout to printf
switch from cin to scanf
switch from dynamic memory allocation to
↪  static memory allocation
The corresponding fast code is:
[FAST CODE]
```

```cpp
#include<bits/stdc++.h>

using namespace std;

int n,k;
long long dp[100005];
long long arr[100005];

int main(){
  scanf("%d%d",&n,&k);
  for(int i=1;i<=n;i++)
  ↪  scanf("%lld",&arr[i]);
  for(int i=2;i<=n;i++){
    dp[i]=1e18;
    for(int j=max(i-k,1);j<i;j++){
      dp[i]=min(dp[i],
      ↪  dp[j]+abs(arr[i]-arr[j]));
    }
  }
  printf("%lld",dp[n]);
}
```

```
[/FAST CODE]
```

Figure 6: Performance optimization example 2.

## C.2. Decompilation

We show several examples to demonstrate how EDITLORD improves the readability of machine-decompiled code.

Given machine decompiled code

```
void func0(char *param_1,long
↪  param_2)
{
  char cVar1;
  long lVar2;
  ulong uVar3;
  char *pcVar4;
  int iVar5;
  bool bVar6;

  lVar2 = -1;
  pcVar4 = param_1;
  do {
    if (lVar2 == 0) break;
    lVar2 = lVar2 + -1;
    cVar1 = *pcVar4;
    pcVar4 = pcVar4 + 1;
  } while (cVar1 != '\0');
  iVar5 = ~(uint)lVar2 - 1;
  if (0 < iVar5) {
    uVar3 = 0;
    do {
      cVar1 = param_1[uVar3];
      if ((byte)(cVar1 + 0x9fU) <
      ↪  0x1a) {
        cVar1 = cVar1 + -0x20;
      }
      else if ((byte)(cVar1 + 0xbfU)
      ↪  < 0x1a) {
        cVar1 = cVar1 + ' ';
      }
      *(char *)(param_2 + uVar3) =
      ↪  cVar1;
      bVar6 = uVar3 != ~(uint)lVar2 -
      ↪  2;
      uVar3 = uVar3 + 1;
    } while (bVar6);
  }
  *(undefined *)(param_2 + iVar5) =
  ↪  0;
  return;
}
```

EDITLORD output

```
The given code describe the following
↪  problem: Convert a string from one case
↪  (upper or lower) to another, handling
↪  special characters appropriately.
The input specification is: The programs take
↪  a string as input, which may contain
↪  uppercase and lowercase letters, and a
↪  buffer to store the converted string.
The output specification is: Output the
↪  converted string, where all letters are
↪  converted to lowercase, while preserving
↪  the case of special characters.
Following editing rules should be applied:
switch from arbitrary pointer checks to
↪  clearer termination conditions
switch from complex pointer arithmetic to
↪  clear variable assignments
switch from indirect variable assignments to
↪  direct return statements
switch from complex multi-step declarations
↪  to clear single declarations
switch from complex logic to simple loops
switch from complex complex if-else structure
↪  to simplified conditional logic
switch from implicit return values to
↪  explicit return values
switch from cryptic variable names to
↪  descriptive variable names
The corresponding source code is:
[ORIGINAL SOURCE CODE]
static void
func0 (char *str, char *buf)
{
  int i, len = strlen (str);
  for (i = 0; i < len; i++) {
    if (str[i] >= 'A' && str[i] <= 'Z')
      buf[i] = str[i] + ' ';
    else if (str[i] >= 'a' && str[i] <= 'z')
      buf[i] = str[i] - ' ';
    else
      buf[i] = str[i];
  }
  buf[len] = '\0';
}
[/ORIGINAL SOURCE CODE]
```

Figure 7: Decomilation example 1.

Given machine decompiled code

```
undefined8 func0(char *param_1,char
↪   *param_2)
{
  char cVar1;
  int iVar2;
  uint uVar3;
  char *pcVar4;
  char *pcVar5;
  long lVar6;
  size_t __n;
  bool bVar7;

  lVar6 = -1;
  pcVar4 = param_2;
  do {
    if (lVar6 == 0) break;
    lVar6 = lVar6 + -1;
    cVar1 = *pcVar4;
    pcVar4 = pcVar4 + 1;
  } while (cVar1 != '\0');
  uVar3 = ~(uint)lVar6;
  iVar2 = uVar3 - 1;
  pcVar4 = (char
↪   *)malloc((long)(int)(uVar3 +
↪   iVar2));
  if (0 < iVar2) {
    lVar6 = (long)iVar2;
    __n = 0;
    do {
      strncpy(pcVar4,param_2 +
↪       __n,lVar6 - __n);
      strncpy(pcVar4 + (lVar6 -
↪       __n),param_2,__n);
      pcVar4[lVar6] = '\0';
      pcVar5 =
↪       strstr(param_1,pcVar4);
      if (pcVar5 != (char *)0x0) {
        free(pcVar4);
        return 1;
      }
      bVar7 = __n != uVar3 - 2;
      __n = __n + 1;
    } while (bVar7);
  }
  free(pcVar4);
  return 0;
}
```

EDITLORD output

```
The given code describe the following
↪   problem: Determine if a string is a
↪   rotation of another string by checking if
↪   any substring of the second string is a
↪   substring of the first string.
The input specification is: The programs take
↪   two strings as input: the first string is
↪   the original string, and the second
↪   string is the potential rotation.
The output specification is: Return a boolean
↪   value indicating whether the second
↪   string is a rotation of the first string.
Following editing rules should be applied:
switch from ambiguous function signatures to
↪   clear function signatures
switch from complex pointer arithmetic to
↪   clear variable assignments
switch from indirect variable assignments to
↪   direct return statements
switch from complex logic to simple loops
switch from complex loops to clear
↪   single-responsibility sections
switch from complex nested conditionals to
↪   clearer sequential logic
switch from complex variable operations to
↪   direct assignments
switch from redundant checks to
↪   straightforward boolean comparisons
The corresponding source code is:
[ORIGINAL SOURCE CODE]
static int
func0(const char *s1, const char *s2)
{
  char *s;
  size_t len, i;
  len = strlen(s2);
  s = malloc(len * 2 + 1);
  if (s == NULL) return (0);
  for (i = 0; i < len; i++) {
    strncpy(s, s2 + i, len - i);
    strncpy(s + len - i, s2, i);
    s[len] = '\0';
    if (strstr(s1, s) != NULL) {
      free(s);
      return (1);
    }
  }
  free(s);
  return (0);
}
[/ORIGINAL SOURCE CODE]
```

Figure 8: Decomilation example 2.

## C.3. Security Hardening

We show several examples to demonstrate how EDITLORD hardens the vulnerable code.

Given vulnerable code

```c
#include <stdlib.h>
#include <string.h>

char*
  allocate_buffer_and_write(size_t
  buf_size) {
    char* buf =
      (char*)malloc(buf_size);
    memset(buf, 'A', 10);
    memset(buf + buf_size - 10, 'A',
      10);
    return buf;
}
```

EDITLORD output

```
The program defines a function that allocates
  a buffer of a specified size and
  initializes its contents by writing 'A'
  characters at both the start and end of
  the buffer. The goal is to ensure that
  the buffer is filled correctly without
  exceeding its allocated size, thereby
  preventing potential buffer overflows.
The input specification is: The function
  takes a single input parameter, buf_size,
  which specifies the size of the buffer to
  be allocated. This size determines how
  much memory will be reserved for the
  buffer.
The output specification is: The function
  returns a pointer to the allocated and
  initialized buffer. If the allocation is
  successful, the buffer will contain 'A'
  characters at the specified positions;
  otherwise, it may return NULL if memory
  allocation fails.
Following editing rules should be applied:
switch from no checks on function return
  values to check function return value
The corresponding secure code is:
[SECURE CODE]
```

```c
char* allocate_buffer_and_write(size_t
  buf_size) {
    char* buf = (char*)malloc(buf_size);
    if (!buf) {
        return 0;
    }
    memset(buf, 'A', 10);
    memset(buf + buf_size - 10, 'A', 10);
    return buf;
}
```
```
[/SECURE CODE]
```

Figure 9: Secure hardening example 1.

Given vulnerable code

```c
#include <stdio.h>
#include <stdlib.h>
#include <sqlite3.h>
#include <string.h>

char *get_user(sqlite3 *db, const
↪   char *username, const char
↪   *password) {
    sqlite3_stmt *stmt;
    char query[256];
    char *userInfo = NULL;

    snprintf(query, sizeof(query),
    ↪   "SELECT userInfo FROM users
    ↪   WHERE username = '%s' AND
    ↪   password = '%s'", username,
    ↪   password);

    if (sqlite3_prepare_v2(db, query,
    ↪   -1, &stmt, NULL) !=
    ↪   SQLITE_OK) {
        fprintf(stderr, "Failed to
        ↪   prepare statement: %s\n",
        ↪   sqlite3_errmsg(db));
        return NULL;
    }

    if (sqlite3_step(stmt) ==
    ↪   SQLITE_ROW) {
        const unsigned char *result
        ↪   =
        ↪   sqlite3_column_text(stmt,
        ↪   0);
        userInfo = strdup((const char
        ↪   *)result);
    }

    sqlite3_finalize(stmt);
    return userInfo;
}
```

EDITLORD output

```
The program aims to retrieve user information
↪   from a SQLite database based on a
↪   provided username and password. It
↪   connects to the database, executes a
↪   query to find the user, and returns the
↪   user's information if found. The program
↪   ensures that the database interactions
↪   are secure and that user input is handled
↪   appropriately to prevent SQL injection
↪   vulnerabilities.
The input specification is: The program takes
↪   three command-line arguments: the
↪   database filename, a username, and a
↪   password. These inputs are used to query
↪   the database for matching user
↪   credentials.
The output specification is: The program
↪   outputs the user's information if a
↪   matching user is found in the database,
↪   or an error message if no user matches
↪   the provided credentials or if an error
↪   occurs during database operations.
Following editing rules should be applied:
switch from direct SQL string interpolation
↪   to use of parameterized logic
The corresponding secure code is:
[SECURE CODE]
```

```c
char *get_user(sqlite3 *db, const char
↪   *username, const char *password) {
  sqlite3_stmt *stmt;
  char *userInfo = NULL;
  const char *query = "SELECT userInfo FROM
  ↪   users WHERE username = ? AND password =
  ↪   ?";
  if (sqlite3_prepare_v2(db, query, -1,
  ↪   &stmt, NULL) != SQLITE_OK) {
    fprintf(stderr, "Failed to prepare
    ↪   statement: %s\n",
    ↪   sqlite3_errmsg(db));
    return NULL;
  }
  sqlite3_bind_text(stmt, 1, username, -1,
  ↪   SQLITE_STATIC);
  sqlite3_bind_text(stmt, 2, password, -1,
  ↪   SQLITE_STATIC);
  if (sqlite3_step(stmt) == SQLITE_ROW){
    const unsigned char *result =
    ↪   sqlite3_column_text(stmt, 0);
    userInfo = strdup((const char *)result);
  }
  sqlite3_finalize(stmt);
  return userInfo;
}
```

```
[/SECURE CODE]
```

Figure 10: Secure hardening example 2.

Given vulnerable code

```c
#include <string.h>

void trim_spaces(char* str) {
    int len = strlen(str);
    while (str[len - 1] == ' ') {
        str[len - 1] = '\0';
        len--;
    }
}
```

EDITLORD output

```
The program defines a function that trims
↪  trailing spaces from a given string. It
↪  modifies the string in place, ensuring
↪  that any whitespace characters at the end
↪  are removed, leaving the rest of the
↪  string intact.
The input specification is: The function
↪  takes a single string input, which is a
↪  character array that may contain trailing
↪  spaces that need to be removed.
The output specification is: The function
↪  does not return any value; instead, it
↪  modifies the input string by removing any
↪  trailing spaces, effectively shortening
↪  the string if necessary.
Following editing rules should be applied:
switch from unvalidated input handling to
↪  check for NULL pointers before access
switch from unvalidated input handling to
↪  check for buffer overflows on memory
↪  accesses
The corresponding secure code is:
[SECURE CODE]
```

```c
void trim_spaces(char* str) {
    int len = strlen(str);
    while (len > 0 && str[len - 1] == ' ') {
        str[len - 1] = '\0';
        len--;
    }
}
```

```
[/SECURE CODE]
```

Figure 11: Secure hardening example 3.

Given vulnerable code

```c
#include <stdio.h>
#include <stdlib.h>

char* read_stdin_to_buffer(int
↪  buf_size) {
    char* buf =
    ↪  (char*)malloc(buf_size);
    gets(buf);
    return buf;
}
```

EDITLORD output

```
The program aims to read a line of input from
↪  the standard input (stdin) into a
↪  dynamically allocated buffer, ensuring
↪  that the input does not exceed the
↪  specified buffer size. It is designed to
↪  prevent buffer overflow vulnerabilities
↪  by limiting the number of characters read
↪  based on the buffer's capacity.
The input specification is: The function
↪  takes an integer input representing the
↪  maximum size of the buffer, which
↪  dictates how many characters can be
↪  safely read from stdin.
The output specification is: The function
↪  returns a pointer to the buffer
↪  containing the user input, ensuring that
↪  the input is properly null-terminated and
↪  does not exceed the allocated buffer
↪  size.
Following editing rules should be applied:
switch from unvalidated input handling to
↪  check for buffer overflows on memory
↪  accesses
switch from no error handling to check for
↪  NULL pointers after memory allocation
The corresponding secure code is:
[SECURE CODE]
```

```c
char* read_stdin_to_buffer(int buf_size) {
    char* buf = (char*)malloc(buf_size);

    fgets(buf, buf_size, stdin);

    return buf;
}
```

```
[/SECURE CODE]
```

Figure 12: Secure hardening example 4.

Given vulnerable code

```
#include <stdlib.h>

int get_price_at_index(int* prices,
↪   int size, char* index_str) {
    int index = atoi(index_str);
    return prices[index];
}
```

EDITLORD output

```
The program defines a function that retrieves
↪   the price of an item at a specified index
↪   from an array of prices. The index is
↪   provided as a string, which is converted
↪   to an integer. The function returns the
↪   price at the corresponding index if it's
↪   valid, or an error if the index is out of
↪   bounds.
The input specification is: The function
↪   takes three inputs: an array of integers
↪   representing prices, the size of the
↪   array, and a string representing the
↪   index of the price to retrieve.
The output specification is: The function
↪   returns the price at the specified index
↪   if it is valid; otherwise, it returns an
↪   error message indicating that the index
↪   is out of bounds.
Following editing rules should be applied:
switch from direct parsing to implement
↪   comprehensive validation checks for
↪   character handling
The corresponding secure code is:
[SECURE CODE]

int get_price_at_index(int* prices, int size,
↪   char* index_str) {
    int index = atoi(index_str);
    if (index < 0 || index >= size) {
        return -1; // Error: index out of
        ↪   bounds
    }
    return prices[index];
}
[/SECURE CODE]
```

Figure 13: Secure hardening example 5.

Table 6: Evaluating DeepSeek-Coder 1.3B on the Code Polish task in CodeEditorBench.

|  | Accuracy | OptScoreTime | OptScore |
|---|---|---|---|
| Finetuned | 0.9% | 0.03% | 0.09% |
| EDITLORD (Ours) | **23.4%** | **1.83%** | **1.19%** |

Table 7: Unseen CWEs evaluation. We also include evaluation on seen CWEs.

|  |  | Correct@$k$ ↑ | | | Security@$k$ ↑ | | | Correct & Sec@$k$ ↑ | | |
|---|---|---|---|---|---|---|---|---|---|---|
|  |  | $k=1$ | $k=10$ | $k=50$ | $k=1$ | $k=10$ | $k=50$ | $k=1$ | $k=10$ | $k=50$ |
| Seen CWEs | Finetuned | 24.3 | 38.4 | 41.7 | **12.8** | **44.0** | 50.0 | 7.7 | 21.6 | **25.0** |
|  | EDITLORD (Ours) | **36.8** | **53.3** | **66.7** | 12.5 | 43.3 | **58.3** | **8.7** | **24.7** | **25.0** |
| Unseen CWEs | Finetuned | 24.1 | 35.0 | 40.0 | 8.6 | 14.8 | 22.5 | 4.6 | 11.5 | 17.5 |
|  | EDITLORD (Ours) | **29.6** | **48.5** | **57.5** | **12.1** | **23.8** | **30.0** | **7.0** | **16.9** | **22.5** |

Table 8: Unseen programming languages evaluation. We also include evaluation on seen language.

|  |  | Accuracy | OptScoreTime | OptScore |
|---|---|---|---|---|
| Seen Language (cpp) | Finetuned | 1.4% | 0.02% | 0.24% |
|  | EDITLORD (Ours) | **28.3%** | **3.1%** | **2.29%** |
| Unseen Languages | Finetuned | 0.7% | 0.04% | 0.02% |
|  | EDITLORD (Ours) | **20.9%** | **1.18%** | **0.63%** |

# D. Rule Learning Details

Given the meta-rule set $G = \{r | r = \text{pre construct} \rightarrow \text{post construct}\}$ (Figure 2), we can manifest these meta-rules to each training sample for augmentation. However, directly prompting the LM to infer meta-rules for each pre-edit and post-edit code pair $(x_i, y_i)$ often leads to hallucinations. To mitigate this, we propose a two-step approach. First, we apply the LM to extract pre construct rules from $x_i$ and post construct rules from $y_i$ separately. Then, we identify which of these pre/post construct combinations are included in the meta rule set. We adopt these verified combined rules as the meta-rules for the code pair $(x_i, y_i)$.

# E. Hyperparameters

To finetune DeepSeek-Coder, we use a default batch size of 32, a learning rate of 1e-5, and 4,000 context lengths for both the input and output tokens. The models are optimized using AdamW and trained for a fixed number of 10 epochs, and we use the model checkpoint that achieves the best validation loss for inference. To finetune GPT-4o mini, we train for only one epoch. At the inference stage, we set the temperature to 0.7 and use the model's default window size, i.e., 16K for DeepSeek-Coder and 128K for GPT-4o-mini.

# F. Additional Experiments

## F.1. Extra Well-Known Benchmark

We further evaluated our finetuned DeepSeek-Coder 1.3B on the Code Polish task in CodeEditorBench (Guo et al., 2025). We follow their metrics by focusing on 1) accuracy: the percentage of problems with correct edits; 2) OptScoreTime: the execution time improvement; and 3) OptScore, the improvement computed by the averaged time and memory. As illustrated in Table 6, EDITLORD, even without extra finetuning on this dataset, outperforms the baseline by 22.5%, 1.8%, and 1.1%, respectively.

Table 9: Comparing EDITLORD to the finetuned baseline under varying amounts of training data.

| | Data Usage | Compile↑ | Correct↑ | Readability↑ | | |
| --- | --- | --- | --- | --- | --- | --- |
| | | | | char | token | emb |
| Robustness | | | | | | |
| Finetuned | 50% | 38.9 | 77.1 | 36.6 | 40.8 | 37.5 |
| EDITLORD (Ours) | 50% | 41.2 | **93.1** | 42.6 | 46.3 | 41.4 |
| | 100% | **46.6** | **93.1** | **44.0** | **47.6** | **41.4** |

## F.2. Out-of-Domain Generalization

**Unseen CWEs.** As described in Section 3.1, our training comes from SVEN (He & Vechev, 2023), but our testing is from CWEval (Peng et al., 2025a) with unseen CWEs. We further analyze the performance of the baseline and EDITLORD on unseen CWEs. As shown in Table 7, EDITLORD generalizes better on unseen CWEs, outperforming the baseline by 38.1%.

**Unseen languages.** We also investigate EDITLORD's generalizability to unseen languages (Python/Java) in performance optimization tasks in CodeEditorBench (Guo et al., 2025) when training on C++ code only. Specifically, we train on the HQ dataset from Shypula et al. (2024), which contains only C++ samples. Table 8 demonstrates that EDITLORD maintains strong generalization to unseen languages, outperforming the baseline by 20.2% in accuracy, 1.14% in execution time improvement, and 0.61% in combined time and memory efficiency improvement.

## F.3. Data Efficiency

To evaluate how EDITLORD scale with varying amounts of finetuning data, we conduct an additional experiment on the decompilation task using only 50% of the finetuning dataset. As shown in Table 9, EDITLORD, trained with just 50% of the data, still surpasses the baseline trained on the full dataset by 5.9%. Moreover, EDITLORD achieves significantly better readability, outperforming the baseline by 13.5% and 16.4% on character- and token-level readability metrics, respectively. This highlights the sample efficiency of EDITLORD, requiring less than 50% of training samples while achieving comparable performance.

