# OpenReview forum: "EditLord: Learning Code Transformation Rules for Code Editing"
_ICML.cc/2025/Conference — ICML 2025 poster_

### Official Review · Reviewer_epTy · 2025-03-09

**Overall Recommendation:** 3

**Summary:**

This paper seeks to decompose the traditional end-to-end LLM-assisted code editing task into discrete and step-wise processes. To this purpose, this paper adopted LLM to summarize meta editing rules from 3 editing tasks: optimization, decompilation and security hardening, and augmented LLM performance in a retrieval style. Their proposed approach achieved SOTA performance in the 3 aforementioned tasks.

**Claims And Evidence:**

The claim of “The post-edit code must be semantically equivalent to the pre-edit code, i.e., having the same functionality, and possessing the desired new properties” raises concerns about the proportion of such edits in real-world scenarios.

**Essential References Not Discussed:**

[1] Liu, Chenyan, et al. "CoEdPilot: Recommending Code Edits with Learned Prior Edit Relevance, Project-wise Awareness, and Interactive Nature." Proceedings of the 33rd ACM SIGSOFT International Symposium on Software Testing and Analysis. 2024.

[2] Priyanshu Gupta, Avishree Khare, Yasharth Bajpai, Saikat Chakraborty, Sumit Gulwani, Aditya Kanade, Arjun Radhakrishna, Gustavo Soares, and Ashish Tiwari. 2023. Grace: Language Models Meet Code Edits. FSE

[3] CODIT: Code Editing With Tree-Based Neural Models. IEEE Transactions on Software Engineering 48, 4 (2022), 1385–1399.

**Experimental Designs Or Analyses:**

Yes, all experimental designs.

**Methods And Evaluation Criteria:**

Code editing tasks are usually treated as translation tasks, and the idea of transforming such an abstract process into a chain-of-edit process is interesting and promising. Meanwhile, the writing style of this paper is easy to follow. However, I have some concerns about this work.

The first concern is the limited scope of the implementation, as it focuses only on three tasks: optimization, decompilation, and security hardening. My intuition is that these tasks are not highly frequent in real-world editing scenarios, which may limit the broader impact of this approach. I would suggest conducting an empirical study to quantify the actual proportion of such edits in real-world settings, providing a clearer picture of the practical significance and applicability of this work.

The next concern is that the lack of quality assurance of the rule set or edit.  Despite the manual inspection, the quality of these rules has not been formally verified. For example, it remains unclear whether the retrieved rules can always be correctly applied to the samples, consistently leading to accurate edits.

To summarize, I encourage the authors to extend the editing scenario to project-wise editing, which holds greater practical significance in software development. Additionally, incorporating methods such as symbolic reasoning to formally verify the rules and assess their applicability would enhance the reliability and robustness of the approach.

**Other Comments Or Suggestions:**

NA

**Other Strengths And Weaknesses:**

NA

**Questions For Authors:**

In practical software engineering, people apply code edits in a repository. In addition, the code edits are triggered by an *issue* (see github issue). How this research can push forward this direction.

**Relation To Broader Scientific Literature:**

NA

**Theoretical Claims:**

NA, there is no theorical claim in the submission.

---

> ### Author Rebuttal · Authors · 2025-04-01
>
> We really appreciate your time and effort in leaving constructive comments.
>
> **Q1: The three tasks it focuses on may limit real-world applicability because they are not highly frequent in real-world editing scenarios.**
>
> The three editing tasks we considered are extensively studied in the literature [1-9], even evaluated on the same datasets as we do. They are broadly applicable already. For example, Scalene, the popular Python profiler, has already integrated GPT to suggest code optimizations for short code snippet [11]; LLMs have been integrated into existing decompilers as plugins [8,9]; Large companies are actively hosting competitions on LLMs to produce secure code [10]
>
> [1] Tan, Hanzhuo, et al. "LLM4Decompile: Decompiling Binary Code with Large Language Models." Proceedings of the 2024 Conference on Empirical Methods in Natural Language Processing. 2024.
>
> [2] Hu, Peiwei, Ruigang Liang, and Kai Chen. "Degpt: Optimizing decompiler output with llm." Proceedings 2024 Network and Distributed System Security Symposium. Vol. 267622140. 2024.
>
> [3] He, Jingxuan, and Martin Vechev. "Large language models for code: Security hardening and adversarial testing." Proceedings of the 2023 ACM SIGSAC Conference on Computer and Communications Security. 2023.
>
> [4] Peng, Jinjun, et al. "CWEval: Outcome-driven Evaluation on Functionality and Security of LLM Code Generation." arXiv preprint arXiv:2501.08200 (2025).
>
> [5] Shypula, Alexander, et al. "Learning Performance-Improving Code Edits." ICLR. 2024.
>
> [6] Huang, Dong, et al. "Effilearner: Enhancing efficiency of generated code via self-optimization." Advances in Neural Information Processing Systems 37 (2024): 84482-84522.
>
> [7] Peng, Yun, et al. "PerfCodeGen: Improving Performance of LLM Generated Code with Execution Feedback." arXiv preprint arXiv:2412.03578 (2024).
>
> [8] aiDAPal: IDA Pro plugin that uses a locally running LLM that has been fine-tuned for Hex-Rays pseudocode to assist with code analysis, https://github.com/atredispartners/aidapal
>
> [9] GhidraAssist: An LLM extension for Ghidra to enable AI assistance in RE, https://github.com/jtang613/GhidrAssist
>
> [10] Amazon Nova AI Challenge accelerating the field of generative AI (LLM for secure code generation), https://www.amazon.science/amazon-nova-ai-challenge-accelerating-the-field-of-generative-ai
>
> [11] Scalene: a high-performance, high-precision CPU, GPU, and memory profiler for Python with AI-powered optimization proposals, https://github.com/plasma-umass/scalene
>
> **Q2: Despite the manual inspection, the quality of these rules has not been formally verified.**
>
> This is a valid concern. While there is no formal guarantee on the generated editing rules’ correctness, our evaluation shows our rule set consistently improves editing performance (Sec. 3.2). We included additional results to show that our meta-rule set learning algorithm can bring improved robustness against the perturbations of the meta-rule set (see below). We do agree that further interacting with formal verifiers to guarantee the correctness of editing rules can be an exciting future work.
>
> ||Correct|Compile|Readability|||
> |:-|-|-|-|:-|:-|
> ||||char|token|emb|
> |EditLord|93.1|46.6|44.0|47.6|41.4|
> |w/ shuffle|94.0|47.3|43.1|45.2|39.7|
>
> **Q3: In practical software engineering, people apply code edits in a repository. In addition, the code edits are triggered by an issue (see GitHub issue). How can this research push forward in this direction?**
>
> EditLord focuses on file-level local edits rather than repo-level editing. File-level edits are already valuable in many scenarios during software development. For example, the developers frequently refactor their code at the local context, e.g., by editing the function they have just written to make it more efficient. While we do believe repository-level editing is indeed an exciting work, we do not attempt to overclaim that EditLord is readily useful for repository-level edits. Instead, we would like to emphasize that the tasks we considered in this paper are already nontrivial and useful, as we have also argued in our response to Q1.
>
> **Q4: Essential references not discussed.**
>
> Thanks for pointing out these related works. We found the main difference between EditLord and many existing editing works (e.g., CoEdPilot, Grace, and CODIT) is that these works mainly focus on repairing functionality, while our tasks focus on preserving original functionality while introducing changes in other dimensions, e.g., performance and readability. That said, these are all very related and should be discussed. We have added discussions in the related works of these papers in our draft.

---

> > ### Comment · Reviewer_epTy · 2025-04-03
> >
> > I thank the authors' response, which generally address my concern. In this case, I would happy to support its acceptance. In the revision, the authors could consider how to extend their file-level editing solutions to a repository-level solution.

---

> > > ### Author Response · Authors · 2025-04-04
> > >
> > > We sincerely thank the reviewer for carefully reviewing our response. We really appreciate your recognition of our effort to address the concerns raised, and we are grateful for your support for the paper's acceptance. We will include a study on EditLord for repo-level code editing tasks, e.g., SWE-Bench, by replacing the default code agent for these tasks with EditLord.

---

### Official Review · Reviewer_QPpP · 2025-03-14

**Overall Recommendation:** 2

**Summary:**

The paper introduces a method for editing code in a decompositional way, where it extracts editing steps, obtains functional specifications, and performs rule-based code editing by prompting LMs.

**Claims And Evidence:**

The paper claims that their method improves code efficiency by leveraging the decompositional nature of code editing tasks. However, after reading the full paper, I don’t fully understand why the extracted editing rules are effective in improving code efficiency or what the rules actually look like. Additionally, their figures need significant improvement, as they don’t help in understanding their methodology—such as the input and output of each subtask, or how the rules and editing process work—but instead confuse me.

I also suggest the authors clearly indicate which dataset benchmarks they use in each table of results. From their experiments section, I gathered that they evaluated on one dataset benchmark, the HQ dataset. However, they also mention the HumanEval dataset elsewhere in the paper, which causes confusion.

Overall, the paper’s presentation requires major revisions and improvements to help readers understand how their method works, and more dataset benchmarks should be included for better evaluation.

**Essential References Not Discussed:**

Not to my knowledge.

**Experimental Designs Or Analyses:**

The number of compared baselines are enough, but more datasets benchmarks should be included for a more comprehensive comparison.

**Methods And Evaluation Criteria:**

See "Claims And Evidence"

**Other Comments Or Suggestions:**

See "Claims And Evidence"

**Other Strengths And Weaknesses:**

See "Claims And Evidence"

**Questions For Authors:**

See "Claims And Evidence"

**Relation To Broader Scientific Literature:**

N/A

**Theoretical Claims:**

Theoretical claims are not required for this work.

---

> ### Author Rebuttal · Authors · 2025-04-01
>
> Thank you so much for your taking the time and effort to read our paper and leave constructive comments.
>
> **Q1. Why extracted editing rules are effective in improving code efficiency or what the rules actually look like.**
>
> These extracted editing rules serve as editing guidelines that help the model reason about the useful steps and actions to take to generate the edited code. Please refer to Table 8 in Appendix B for some examples of the rules. We have also added the detailed samples to the [anonymous repo](https://anonymous.4open.science/r/EditLord-9C9B/example.pdf).
>
> **Q2. The figures are confusing and do not clearly illustrate the methodology, e.g., the input/output of each subtasks, rule usage, and the editing process.**
>
> We apologize for the potentially unclear figure presentation. To clarify, we take the performance optimization task as an example in Fig 1. The input is a slow (pre-edit) code, while the expected output is semantically equivalent but faster (post-edit) code. Now consider inferencing by an EditLord finetuned model. Instead of directly generating the faster code (post-edit code), the model will first generate functional specifications and editing rules, and then these generated is in the context, and the model keeps on generating the post-edit (faster) code.
>
> For the decompilation task, the input is an unreadable, non-idiomatic ghidra-decompiled (pre-edit) code, while the expected output is a more readable, idiomatic (post-edit) code (see Fig 3).
>
> For the security hardening task, the input is vulnerable code, while the expected output is secure code.
>
> In the upper part of Fig 2, we illustrate how we prepare the functional specifications and editing rules. We will take the pre-edit and post-edit code pair from the training data as input and use LLMs to annotate the functional specification and the editing rules for each pair.
>
> Since editing rules are often shared among different samples, our Alg. 1 describes how an LM serves as a rule learner by iteratively refining the raw meta-rule set and producing a more concise meta-rule set by using operations ADD, MERGE, and PRUNE. This step is done *independently* for each task.
> We also state the input and output for each editing mode more clearly in the [updated figure](https://anonymous.4open.science/r/EditLord-9C9B/workflow.pdf) for our improved workflow figure.
>
> **Q3. The dataset used in their experiment is unclear and causes confusion (e.g., HQ, HumanEval).**
>
> We evaluate different editing tasks with different datasets, as different objectives require different evaluations. Specifically, for performance optimization tasks, we train on the HQ dataset and evaluate on the test split from PIE. For the decompilation task, we train on AnghaBench and test on HumanEval-Decompile. For the security hardening task, we train on SVEN, and test on CWEval. We will update our Section 3.1 to describe the setting more clearly.
>
> **Q4. More benchmarks should be included for better evaluation.**
>
> Thanks for pointing this out. Per our response to reviewer raGi, we evaluated EditLord’s finetuned DeepSeek-Coder 1.3B in CodeEditorBench, the benchmark we did not consider in the submission. We follow their metrics by focusing on 1) accuracy: the percentage of problems with correct edits; 2) OptScoreTime: the execution time improvement; and 3) OptScore, the improvement computed by the averaged time and memory together.
>
> Surprisingly, without further finetuning EditLord on this benchmark, but simply running inference using a finetuned EditLord model on a completely different training set, it substantially outperforms the finetuned baseline by 22.5%, 1.8%, and 1.1%, respectively.
>
> ||Accuracy|OptScoreTime|OptScore|
> |:-|:-|:-|:-|
> |Finetuned|0.9%|0.03%|0.09%|
> |EditLord|23.4%|1.83%|1.19%|
>
>
> **Q5. Repo seems empty.**
>
> Thanks for your interest in our artifact. We would hope to clarify that the repository is not empty. The code and prompts were uploaded along with the paper submission. It is likely that the README is blank, which makes the repo appear to be incomplete. We have already uploaded other important files and updated the README. We hope this addresses your concern.

---

### Official Review · Reviewer_wc39 · 2025-03-14

**Overall Recommendation:** 4

**Summary:**

EditLord is a system designed to improve performance on code editing tasks (e.g. performance/readability/security). The proposed pipeline involves a few steps, starting from a training dataset (editing task-specific) with pre/post edit programs. In the first step, a LLM is used to produce a set of editing ‘meta-rules’. In the second step, a LLM is used to describe the transformation that takes places for each training task. Experiments show, across different tasks, that finetuning a LLM with these extra annotations leads to better performance versus finetuning a LLM on just the tasks alone.

**Claims And Evidence:**

Yes.

**Essential References Not Discussed:**

References largely seem appropriate.

Although not 'essential', I would encourage the authors to add a discussion of works that try to discover similar artifacts as their 'meta rule-sets'. E.g. for program synthesis (not editing) library learning works (DreamCoder, LILO), or for more general LLM tasks skill discovery (TroVE, GENOME). This connection seems useful, and relevant, though I think even including it in the appendix would be fine.

**Experimental Designs Or Analyses:**

The experiments seem sound in design.

**Methods And Evaluation Criteria:**

Yes they seem to.

**Other Comments Or Suggestions:**

One thought that I had while reading the paper: the proposed process basically creates better annotations for LLM finetuning. When trained on the entire dataset, these annotations lead to improved performance, but how does this performance improvement change as a function of the amount of finetuning data used? I would almost expect this method to do 'better' (than the default finetuning strategy) when the amount of finetuning data is small (would be a useful appendix/supplemental experiment).

**Other Strengths And Weaknesses:**

This is a strong paper. It offers an interesting solution for a hard, well-studied problem, demonstrating consistent experimental improvement over a range of tasks.

The idea of trying to extract out meta-rules from a corpora of editing tasks is noteworthy and to my knowledge has not be tried before; beyond benefits during finetuning, this also allows for human-in-the-loop improvements through editing / curation of these rule sets (Section 3.6).

The experimental framework seems robust and quite comprehensive; from my read, the paper does a good job of supporting their claims against prior finetuning alternatives.

The most significant weakness of the paper is that its unclear how robust this meta-rule creation process actually is; while it certainly proves consistently useful, the machinery to produce these rules is relatively simple from a certain perspective (which in some ways is a positive of the system). There aren't really any guarantees that these meta-rules will be good / interpretable / general (unless the human-in-the-loop machinery is employed), but this can be left as a problem for future work / investigations.

**Questions For Authors:**

Though not critical, it would have been interesting to see how larger LLMs (not finetuned) would perform on these tasks, especially when the rules/specifications are appended to the prompt or not.

How stable is convergence of meta-rule set? I.e. if you re-order the rules from 2.2 before passing them into Alg 1, do the same types of meta-rules get discovered, or does this have high variance? I think more analysis on both the stability meta-rule set generation process, and how well it matches 'semantics' (human prior, by some measure) would bring the paper from good to great.

Clarification question: What is DIS in algorithm 1? Is this some text-embedding distance?

**Relation To Broader Scientific Literature:**

This system works on the problem of code editing/improvement; offering a general method/strategy for improving LLM finetuning for this task.

**Theoretical Claims:**

N/A

---

> ### Author Rebuttal · Authors · 2025-04-01
>
> We are grateful for your effort in leaving us such encouraging comments.
>
> **Q1. It’s unclear how robust the meta rule creation process is. And there aren’t really any guarantees on the quality without human interaction.**
>
> Great point. We observed that if we simply bootstrap per-sample rules, the rules often end up being noisy, repetitive, and not generic. This leads the resulting meta-rule set to be too large to fit in the model context, and the resulting per-sample manifested rules become extremely susceptible to the order of training samples. This motivated us to *mitigate* this issue by developing the meta-rule learning algorithm to keep the meta-rule set concise and generalizable. As you have also noted, our evaluation in Sec 3.6 shows that exposing the meta-rule set during inference allows us to invite human intervention to further improve the quality of the edited code.
>
> While there is no formal guarantee on the generated editing rules’ correctness, our evaluation shows our rule set consistently improves editing performance (Sec. 3.2). In our response to your Q4, our additional results show that our meta-rule set learning algorithm is relatively robust against the perturbations of the training samples. We do agree that bringing provable guarantees to the editing rules, e.g., by interacting with formal verifiers brought by the compiler techniques, is an exciting future work.
>
> **Q2. It’s unclear how annotation-driven performance improvements scale with varying amounts of finetuning data.**
>
> This is a great suggestion. We have added an experiment on the decompilation task when using only 50% of the finetuning data. The results show that EditLord, with only 50% of the training set, still outperforms the finetuned baseline with 100% training samples by 5.9%. Its readability results remain 13.5% and 16.4% higher than the finetuned baseline on char- and token-level readability. This result demonstrates that the sample efficiency brought by EditLord, requiring less than 50% of training samples while achieving comparable performance.
>
> |Data Usage|DeepSeek-Coder-1.3B|Correct|Compile|Readability|||
> |:-|:-|-|-|-|:-|:-|
> |||||char|token|emb|
> |50%|EditLord|41.2|93.1|42.6|46.3|41.4|
> |100%|Finetuned|38.9|77.1|36.6|40.8|37.5|
> |100%|EditLord|46.6|93.1|44.0|47.6|41.4|
>
> **Q3. What is the performance of larger, non-finetuned LLMs w/ or w/o rules/specifications appended?**
>
> We added an experiment exploring the GPT-4o-mini’s (2024-07-18 version) performance when incorporating our meta-rule set and functional specifications. As shown below, simply including them for non-finetuned GPT-4o-mini will improve its correctness and readability by 11.9% and 8.6%, respectively.
> ||Correct|Compile|Readability|||
> |:-|-|-|-|:-|:-|
> ||||char|token|emb|
> |Prompt|44.3|61.8|33.1|37.0|37.9|
> |w/ EditLord|49.6|57.3|36.2|40.1|41.0|
>
>
> **Q4. How stable is the convergence of the meta-rule set? Will shuffling introduce high variance?**
>
> We have added an experiment that randomly shuffles the initial dataset before passing it into Alg. 1 and obtaining a pair of meta-rule sets. We then measure the similarity between these two sets. Specifically, for each rule in one set, we compute its average semantic similarity to the top 5 most similar rules in another set and average them again to obtain the overall similarity. The semantic similarity between two rules is computed by the cosine distances of the rule embeddings computed by CodeSage, following the similar setting as in the readability metrics in Sec. 3.1. The resulting semantic similarity between the shuffled rule sets is 0.87, close to 1.
>
> We also show the end-to-end results on the robustness of EditLord introduced by the resulting meta-rule set. The perturbed rule sets lead to less than 2.4% performance changes.
> ||Correct|Compile|Readability|||
> |:-|-|-|-|:-|:-|
> ||||char|token|emb|
> |EditLord|93.1|46.6|44.0|47.6|41.4|
> |w/ shuffle|94.0|47.3|43.1|45.2|39.7|
>
> **Q5. What is DIS in Alg. 1?**
>
> DIS is the internal embedding distance calculated by the LLM. We intended to use DIS to describe that we ask the LLMs to discover the rules that share similar semantics to decide whether to directly ADD the current rule to the rule set or MERGE it with an existing rule in the rule set. We acknowledge that this can lead to confusion, and we have updated the algorithm at [link]((https://anonymous.4open.science/r/EditLord-9C9B/algo.pdf)).
>
> **Q6: Essential References Not Discussed.**
>
> Thanks for pointing out these related benchmarks. We have added the discussion to our draft. We view code generation tasks as complementary to ours as their input is not the code, but usually natural language specification, similar to our functional specification.
>
> Thanks for pointing out skill discovery works, e.g., TroVE and GENOME! We were inspired by TroVE in the very early stage of our project but ended up narrowing down our focus to primarily code-editing works. We should definitely discuss both, and we have added them to the draft.

---

### Official Review · Reviewer_raGi · 2025-03-24

**Overall Recommendation:** 4

**Summary:**

Traditionally, for code editing, language models (LMs) are often used to directly generate the output code (or diff) given the input code in a single turn. There have also been approaches that prompt LMs in a CoT-style manner to generate some reasoning before outputting the edited code. Similarly, existing approaches for supervising LMs for code editing focus on directly generating the edited code (optionally augmented with some reasoning).

This paper offers a fresh perspective on supervising LMs for code editing tasks. Instead of supervising an LM to directly transform the input code to the output code, the proposed approach first produces an understanding (functional specification) of the input code, a set of rules to transform the input code, and finally, the output code conditioned on the functional specification and the set of rules.

The paper presents experiments considering 3 different code editing tasks and 3 recent language models (DeepseekCoder-1.3B, DeepSeekCoder-6.7B, and GPT4o-mini). Authors compare their fine-tuning approach with naive fine-tuning; zero-shot prompting; and chain-of-through prompting, for each task and for each model. The results indicate that the proposed fine-tuning approach consistently outperforms the standard fine-tuning and other baselines.

Authors also present ablations demonstrating their approach to be more robust w.r.t. semantics preserving code transformations and length of the input code.

**Claims And Evidence:**

Claims made in the paper are supported by clear and convincing evidence.

However, for further validation of the results, it would be nice to have additional experiments on some other well-known existing code-editing benchmarks. Please see "Essential References Not Discussed".

**Essential References Not Discussed:**

The following code-editing benchmarks are fairly well known and should be discussed and included in experiments

[1] [CodeEditorBench: Evaluating Code Editing Capability of Large Language Models](https://arxiv.org/abs/2404.03543)

[2] [NoFunEval: Funny How Code LMs Falter on Requirements Beyond Functional Correctness](https://arxiv.org/abs/2401.15963)

[3] [Aider Code Editing Benchmark (including aider polyglot)](https://aider.chat/docs/benchmarks.html)

[4] [HumanEvalFix](https://arxiv.org/abs/2308.07124) (a code-editing variant of Human eval)

**Experimental Designs Or Analyses:**

Experimental designs seem sound and valid.

However, for further validation of the results, it would be nice to have additional experiments on some other well-known existing code-editing benchmarks. Please see "Essential References Not Discussed".

**Methods And Evaluation Criteria:**

Proposed methods and/or evaluation criteria make sense for the problem or application at hand.

**Other Comments Or Suggestions:**

Comments
* Post-edited code may not always necessarily be equivalent to the pre-edited code in case of bug fixing. Therefore, s_i should be a function of both x_i and y_i and not x_i alone.
* If possible, the paper should included zero-shot & CoT performance of larger models like GPT4-o as well. It would be interesting to see if fine-tuning GPT4o-mini makes is as good as GPT4-o for code-editing.

**Other Strengths And Weaknesses:**

Strengths

* Paper should serve as an interesting read for audience interesting in language models for code editing
* The paper presents experiments considering 3 different code editing tasks and 3 recent language models (DeepseekCoder-1.3B, DeepSeekCoder-6.7B, and GPT4o-mini). Authors compare their fine-tuning approach with naive fine-tuning; zero-shot prompting; and chain-of-through prompting, for each task and for each model. The results indicate that the proposed fine-tuning approach consistently outperforms the standard fine-tuning and other baselines.
* Authors also present ablations demonstrating their approach to be more robust w.r.t. semantics preserving code transformations and length of the input code.

Weakness
* While experiments are fairly detailed, the paper currently ignores some well-known benchmarks specifically designed for code-editing. Covering these benchmarks should help strengthen the claims in the paper and add more credibility. (this is the only major weakness I observe in this paper.)
* The proposed approach assumes a training dataset in the target domain. (What about covering security vulnuerabilities that were not a part of the training dataset?). Investigating out-of-domain generalization of the proposed approach should be interesting.
* Paper currently does not provide examples corresponding to each of the three datasets. One representative example from each of the three code-editing tasks would be very helpful. It will help us understand the granularity of the input code (function level/ class level/ or file level) for each task.

**Questions For Authors:**

* How robust is the proposed approach to noise in rule manifestations? During training rules are manifested while making use of both x_i and y_i. During testing, we don’t have y_i, so we might end up using irrelevant rules.

* How large is the discovered rule set for different code editing scenarios? Is it possible to share entire rule set for each of the three scenarios/datasets? Additionally, how large is the initial rule set G^0 and the final rule set G?

* Have authors considered extending CoT reasoning with their rule set? Explicitly augmenting CoT reasoning to use rules might give better results?

**Relation To Broader Scientific Literature:**

The key contributions are related and relevant to prior literature on using language models for code editing.

The paper utilizes benchmarks developed by the following prior literature on code editing for evaluating their proposed approach.

[1] Learning performance-improving code edits. arXiv preprint arXiv:2302.07867, 2023.

[2] Llm4decompile:Decompiling binary code with large language models. arXiv preprint arXiv:2403.05286, 2024.

[3] Cweval: Outcome-driven evaluation on functionality and security of llm code generation. arXiv preprint arXiv:2501.08200, 2025.

**Theoretical Claims:**

There are no theoretical claims in the paper as such.

---

> ### Author Rebuttal · Authors · 2025-04-01
>
> We really appreciate your time and effort in reviewing our paper and giving us constructive comments!
>
> **Q1: More well-known benchmarks should be included.**
>
> Thanks for pointing this out. We originally focused on individual tasks where the corresponding papers also proposed tailored solutions (e.g., PIE) to ensure we compare to the state-of-the-art baselines.
>
> That said, we evaluated our finetuned DeepSeek-Coder 1.3B on the Code Polish task in CodeEditorBench. We follow their metrics by focusing on 1) accuracy: the %problems with correct edits; 2) OptScoreTime: the execution time improvement; and 3) OptScore, the improvement computed by the averaged time and memory. EditLord, even without extra finetuning on this dataset, outperforms the finetuned model by 22.5%, 1.8%, and 1.1%, respectively.
>
> ||Accuracy|OptScoreTime|OptScore|
> |:-|:-|:-|:-|
> |Finetuned|0.9%|0.03%|0.09%|
> |EditLord|23.4%|1.83%|1.19%|
>
>
> **Q2: How does EditLord work on out-of-domain generalization (e.g., unseen vulnerabilities)?**
>
>
> We ensured our training and evaluation came from two data sources. As described in Sec 3.1, our training comes from SVEN, but our testing is from CWEval with unseen CWEs.
> We add below a breakdown of the baseline and EditLord’s performance on seen/unseen CWEs. EditLord consistently generalizes better than the baseline, outperforming it by 7.5% and 38.1%, respectively.
> We also add generalization tests on unseen languages (Python/Java) in performance optimization in CodeEditorBench. EditLord achieves improvement in both seen and unseen languages, outperforming it by 2.05% and 0.61%, respectively.
>
> Along with the length generalization results in Sec 3.5, our added experiments here show that EditLord maintains strong generalization in various settings. We will include them in the draft.
>
> |Security|Methods|Correct@k|||Security@k|||Correct & Sec@k|||
> |:-|:-|:-|:-|:-|:-|:-|:-|:-|:-|:-|
> |||k=1|k=10|k=50|k=1|k=10|k=50|k=1|k=10|k=50|
> |Seen CWEs|Finetuned|24.3|38.4|41.7|12.8|44.0|50.0|7.7|21.6|25.0|
> |Seen CWEs|EditLord|36.8|53.3|66.7|12.5|43.3|58.3|8.7|24.7|25.0|
> |Unseen CWEs|Finetuned|24.1|35.0|40.0|8.6|14.8|22.5|4.6|11.5|17.5|
> |Unseen CWEs|EditLord|29.6|48.5|57.5|12.1|23.8|30.0|7.0|16.9|22.5|
>
> |Code Polish|Methods|Accuracy|OptScoreTime|OptScore|
> |:-|:-|:-|:-|:-|
> |Seen lang (cpp)|Finetuned|1.4%|0.02%|0.24%|
> |Seen lang (cpp)|EditLord|28.3%|3.1%|2.29%|
> |Unseen lang|Finetuned|0.7%|0.04%|0.02%|
> |Unseen lang|EditLord|20.9%|1.18%|0.63%|
>
>
> **Q3: The granularity of the input code (function/class/file level) for each task is unclear.**
>
>
> The input code is at the file level for all tasks. This ensures the code can be compiled to measure the functional correctness. Please see detailed examples [here](https://anonymous.4open.science/r/EditLord-9C9B/example.pdf).
>
>
> **Q4: How robust is EditLord to noise in rule manifestations during inference?**
>
>
> Great point. We observed that learning rules for each sample often leads to noisy and repetitive rules, which can degrade performance. This motivated us to propose Alg.1 to disentangle the meta-rule learning and the per-sample rule manifestation. While there is no formal guarantee on the generated editing rules’ correctness, our evaluation shows our rule set consistently improves editing performance (Sec. 3.2). That said, generating provably correct rules in formal languages with formal verifiers is indeed an exciting future work.
>
> We include some preliminary results on the robustness of EditLord against randomly shuffled rules for training. The following shows that perturbed rule sets lead to less than 2.4 performance changes in the decompilation task.
> ||Correct|Compile|Readability|||
> |:-|-|-|-|:-|:-|
> ||||char|token|emb|
> |EditLord|93.1|46.6|44.0|47.6|41.4|
> |w/ shuffle|94.0|47.3|43.1|45.2|39.7|
>
>
> **Q5: How large is the initial rule set G^0, and the final discovered rule set G? Can you share the entire rule set for each task?**
>
>
> The initial rule set $G^0$ has 2.9K, 1.9K, and 1.2K rules for performance, decompilation, and security hardening, respectively, while the final rule set G has 221, 228, 237, respectively. Table 8 shows the example rules. Thanks for your interest. We will definitely release the full set once the paper is ready to be published.
>
>
> **Q6: Will extending CoT reasoning with their rule set give better results?**
>
>
> Yes, we included this study in the paper. As described in Fig.2, the *prompting* editing mode refers to CoT prompting with our meta-rule set. In Fig.4 (R=0 means CoT prompting without iteratively refining the generated code), we compare this CoT setting with zero-shot prompting w/o CoT (i.e., w/ vs w/o EditLord). Including our rules improves the CoT performance by 46% across all the tasks.
>
>
> **Q7: Related work not discussed.**
>
>
> Thanks for pointing out these benchmarks. We have included preliminary results on CodeEditorBench (Q1), showing EditLord’s potential to generalize to unseen benchmarks. We will add new results and discussions of these benchmarks in our paper.

---

### Decision · Program_Chairs · 2025-05-01

**Decision:**

Accept (poster)

**Comment:**

The manuscript proposes a novel approach to improving the code-editing abilities of code language models. The authors propose a simple and intuitive formulation of the editing problem -- as one of infering edit rules given training (before, after) code pairs, and then using the inferred rule sets to improve the editing accuracy of LMs. The proposed solution is elegant and the writing brings out the novelty, technical contributions, as well as the merits of the proposed method.

I am delighted to see that the authors have shown great rigor in their experiment design, ablations, comparisons with competing methods and baselines. I'm also impressed with the experimental validation on metrics/scenarios beyond standard functional correctness, including implementation efficiency and security.

Overall, I believe the paper makes good contributions to an important software engineering problem, has strong technical contributions, and a thorough set of evaluations backing the proposed method. I recommend accept.